# communications
# engineering

# The economic value of augmentative exoskeletons and their assistance

Roberto Leo Medrano[1,2,3], Gray Cortright Thomas[1,2,3,4], Drew Margolin[5] & Elliott J. Rouse [1,2,3✉]

For augmentative exoskeletons that assist able-bodied users, a clear metric of success remains an open question. Here we leverage the Vickrey second-price auction to quantify the economic value added by lower-limb exoskeletons and their assistance. We posited that if exoskeletons provided helpful assistance during a difficult task, this value could be quantified through a lowering of participant auction bids to continue walking. The bidding results were compared across different conditions to determine the economic value of the exoskeleton, bearing in mind also the cost of wearing the added mass of the exoskeleton. Results show that the total value of the exoskeleton and assistance was modest. While most participants found the assistance itself valuable, this value was mostly offset by the extra mass added of wearing the exoskeleton. Our approach provides insight into how exoskeleton wearers may value different aspects of user experience. These results suggest economic value may be a powerful tool in the design and control of exoskeletons that maximize user benefit.

[1] Neurobionics Lab, University of Michigan, Ann Arbor, MI, USA. [2] Mechanical Engineering, University of Michigan, Ann Arbor, MI, USA. [3] Department of Robotics, University of Michigan, Ann Arbor, MI, USA. [4] Electrical Engineering and Computer Science, University of Michigan, Ann Arbor, MI, USA. [5] Communication, Cornell University, Ithaca, NY, USA. ✉email: ejrouse@umich.edu

Powered exoskeletons have long fascinated the public consciousness with their promise to supersede the limitations of human performance. In parallel, there has been substantial growth of scientific research into powered lower-limb exoskeletons, driven by the potential of these technologies to transform mobility by extending the locomotor abilities of their wearers. Powered exoskeletons augment lower-limb function by providing mechanical assistance to the joints of the legs in tandem with the human neuromotor system, and thus can make physically demanding tasks less challenging for both able-bodied and impaired individuals. Assistance provided by modern exoskeletons have been shown to reduce the caloric demands and muscular effort required for walking[1–9]. Consequently, exoskeletons may have beneficial implications for recreational users and workers in factory, military, or supply chain environments. In addition, exoskeletons can reduce muscle activation, and thus certain exoskeleton architectures may reduce joint loading[10–13], potentially extending the physical capabilities of aging individuals. Rehabilitation-focused exoskeletons may also restore the mobility of people with neuromotor deficits who face weakness and impairments in balance, coordination, and joint mechanics following upper motor neuron disease (e.g., stroke). Exoskeletons can be used to assist the gait of these individuals, with several commercially-available lower-limb exoskeletons having been approved by the U.S. Food and Drug Administration.

The metrics by which we assess exoskeletons drive their design, control, and potential impact. For rehabilitative applications, the design of these technologies has a clear physiological objective: the restoration of impaired gait function. Rehabilitative exoskeletons can achieve this goal by directly affecting the kinetics and kinematics via their applied assistance[8,12,14–16] or by using the exoskeleton as a training aid to foster neurorehabilitation[17–19]. Thus, the use of these exoskeletons provides a clear physiological objective on which to base the design and control decisions required for development. However, for augmentative applications in which the user is typically able-bodied, the metrics of success are less clear. Currently, augmentative exoskeletons are developed based on their ability to meet a physiological objective; the "gold standard" for exoskeleton success is the reduction of metabolic expenditure during locomotion (i.e., a reduction of the calories burned)[20]. This objective is both intuitively meaningful and objectively measurable[21,22]. Modern exoskeletons have reduced the metabolic rate relative to unassisted walking by an average of ~14%[2–6,23]. This objective has led to the rise of promising 'human-in-the-loop' (HILO) optimization techniques, which directly modulate exoskeleton assistance based on the metabolic reductions experienced by the wearer[24–30]. Recent work has established experimental infrastructure that illustrates the tight coupling between metrics of success and exoskeleton development. An example of this infrastructure includes tethered emulator systems[7,31], whose purpose is to inform exoskeleton design and control based on their ability to reduce the metabolic expenditure of their wearer. Numerous studies investigating the biomechanical underpinnings of metabolic cost reductions have also been conducted to find more optimal exoskeleton assistance settings[23,25,32,33]. Finally, other metrics include net-joint torque reduction[11] and muscle activation reduction[10,12], which are commonly used, easier-to-measure proxies for improvement in energetics.

Though the physiological benefits of exoskeletons have been demonstrated using metabolic metrics, these benefits have not widely translated to wearer perception of enhanced endurance and strength. These perceptions are important, as for augmentative exoskeletons to reach their potential in society, users will need to voluntarily accept these technologies into their lives. They thus must be developed to provide a perceivable benefit to their wearer, in addition to objective assessment of their impact. Our recent work has shown that during short-term exoskeleton-assisted walking, the average user cannot yet perceive the benefit of most systems available today[34,35]. That is, the metabolic rate needed to be reduced by 23% ($N = 10$) before exoskeleton users could reliably perceive this improvement (whereas most modern exoskeletons reduce the wearer's metabolic rate by 14% compared to unassisted walking[20]). These results agree with prior studies that showed humans were relatively insensitive to small changes in exertion in other exercise contexts[36]. Intuitively, if the user is unable to perceive the metabolic reduction provided by an exoskeleton, this value may be difficult to incorporate into decision-making during exoskeleton design, translation, and adoption. Consequently, assessing and developing exoskeletons based on reductions in metabolic rate could result in systems that are not perceived as valuable by users, despite significant energetic benefits[2,26,33,37–39].

An alternative method for measuring success in exoskeleton development is to quantify the perceived economic value provided to the wearer during use. Economic value, measured in monetary currency (e.g., US Dollars), is assigned by the wearer and can reflect the multifaceted nature of exoskeleton user experience. Although exoskeletons can provide assistance that improves energetics, that assistance often comes at a cost to the wearer. Exoskeletons can add discomfort, weight, and audible noise, in addition to having aesthetic implications. While exoskeletons may potentially have universal positive value, the heterogeneity of the metabolic response to exoskeleton assistance, coupled with the known variety of responses to new innovations within the social sciences[40], could also imply a wide range of valuations for wearing an exoskeleton. If the user is able to assign economic value to the experience of exoskeleton use, they are able to inherently balance and quantify these trade offs. Thus, we posit that exoskeletons that maximize economic value may have a greater likelihood for adoption and use. Prior work in management science has established that the perceived value of different technologies has a significant impact on user intent to adopt those technologies into their daily lives[41–45]. When potential exoskeleton users, manufacturers, and others are weighing the choice to adopt or purchase an exoskeleton, the consciously perceived benefits must outweigh these costs. Thus, the perceived economic value of exoskeleton use is a potentially powerful metric for designing and controlling exoskeletons that quantifies meaningful, individualized benefits to wearers.

In this study, we introduce a tool for measuring the perceived short-term economic value of exoskeleton use as a metric to evaluate their performance and user experience. We define and use this tool to quantify the economic value—termed marginal value (MV)—provided by bilateral ankle exoskeleton use during uphill walking. We leveraged the Vickrey second-price auction to measure participants' "price to walk," across different walking conditions, such as walking with and without exoskeleton assistance. We obtain our economic value metric (i.e., MV) that captures the value of the exoskeleton by calculating the difference in cumulative value between these conditions. Since both cumulative values encode the baseline valuation of walking time, the difference in values is due to the effects of wearing the exoskeleton. Our experiment revealed that while there was insignificant positive value of exoskeleton use across subjects, there was a large disparity between subjects. Some subjects reported substantial value provided by the exoskeletons, which represents an opportunity target these "responders" in future work. The near-zero net MV from the exoskeleton stems from two competing effects: the MV of the powered assistance alone was substantially positive, but it was counteracted by the cost from wearing the device itself. The use of MV offers advantages

over the more common metabolic rate metric, including its accessibility, in that it does not require specialized equipment, and its intuitiveness, as users and manufacturers are more likely to understand the value of monetary currency over biomechanical quantities (e.g., calories burned or muscle power). Our approach is also generalizable, and can be used to measure the value of not only different types of exoskeleton assistance, but also various technologies, activities, or experimental conditions. Thus, this study represents an initial investigation into the use of economic value as a metric to assess the success of exoskeletons and their assistance; we believe assessment of economic value represents a potentially useful alternative to the dominant approach of quantifying the reduction in metabolic rate.

## Results

### Background on the Vickrey auction

The Vickrey auction[46] is a powerful economic tool for determining the true value placed on goods or actions. In this type of auction, participants compete to purchase (or sell) a good or item. For each participant, the auction's structure is designed such that the optimal strategy for obtaining the item is to truthfully represent their internal value with their bid (e.g., to bid an amount equal to the true worth of the item). This optimality stems from the second-price nature of the auction[46–48], in which the winner is the participant who bids the highest (or lowest, as in the selling implementation used in this study). However, rather than paying the highest bid, the winner of the auction instead pays the second-highest bid. This feature—awarding the item to the highest bidder but requiring them to pay only the second-highest bid—addresses a problem present in standard auctions in economic theory, which is that rational bidders will bid not only based on their own valuation of the item (the information the auction aims to reveal) but also their assessment of the other bidders' valuation of the item. Specifically, in a standard auction in which the item is awarded to the highest bidder at the price they bid, bidders have an incentive to bid only slightly above what they think the highest bid from their competitors will be, thus "under-bidding" their true value. The Vickrey auction removes this incentive. Participants do not know the value of competing bids (sealed-bid) before submitting their bids. In theory, bidders are disincentivized to bid less than their true value, as they run the risk of not winning the auction (not acquiring the good), and do not gain by bidding just above the second-highest bid (since they pay that second-highest bid in any event). Similarly, participants should not bid more than their true value, which otherwise could cause them to pay more than the value of the item. Thus, the second-price nature breaks the link between the auction winner and their specific bid. The inverse is also true for the case of selling an item (second-lowest bid is paid, lowest bid wins); thus, the incentive structure that elicits truthful bidding still holds in the seller's auction. Due to the presence of this optimal strategy, the Vickrey auction provides a method for quantifying the value of arbitrary goods, services, or abstract concepts[48,49]. Prior researchers have also used Vickrey auction metrics to measure the value of abstract concepts or actions, such as food safety[50], GMO-free foods[51], the stigma resulting from HIV[52], personally identifiable information[53], and smartphone battery life[54]. In particular, Coursey et al. employed the Vickrey auction to quantify the willingness of participants to endure performing an unpleasant task, such as tasting a bitter liquid[55]. In our protocol, we use the Vickrey auction sequentially to repeatedly sample individuals' valuations of their time during uphill walking. Participants competed in a series of auctions in which they auctioned off their walking time for 2-min intervals in exchange for actual monetary compensation. If the participant won the auction, they accrued the payout and walked for the

2-min interval, whereas if they lost, they did not receive the payout and rested until the next interval. The participants bids across the sequential auctions denotes the wearer's "price-to-walk" curve. Participants walked in different walking conditions, which included normal unassisted walking and walking with exoskeleton augmentation; by comparing the price-to-walk curves for these different conditions, the value of the exoskeleton assistance can be extracted.

### Study results

Within each trial, participant bids invariably trended upwards as participants became more fatigued and this trend was well represented by a first-order exponential. During each trial, participants either walked with normal walking shoes ('walking-no-exo'), with the exoskeleton applying assistance ('exo-powered'), or with the exoskeleton donned but applying no power ('exo-powered-off'). Across all three walking conditions, the average bid for each 2-min interval was \$0.75, with a standard deviation of \$0.39. The maximum bid was \$5.00, while the minimum bid was \$0.10 (the total bids from all participants are shown in Supplementary Figure 1). As participants continued to walk during each trial, their bids increased at varying rates (Fig. 1). The first-order exponential model exhibited an $R^2$ of $0.87 \pm 0.11$, averaged over all conditions experienced by the sixteen participants. These first-order responses denote the user's price to walk curves for each condition (the curves for the walking-no-exo and exo-powered conditions are shown in Fig. 1a). The area between the walking-no-exo and exo-powered curves denotes the Marginal Value (MV) of the exoskeleton and its assistance, which is the value obtained by the participant from the exoskeleton's use. Participant price-to-walk curves broadly demonstrated three potential outcomes; namely, a clear economic benefit from the exoskeleton's assistance (higher walking-no-exo curve than exo-powered curve, Fig. 1b, positive MV), a clear economic penalty (lower walking-no-exo curve than exo-powered curve, Fig. 1c, negative MV), and a negligible economic effect (similar walking-no-exo and exo-powered curves, Fig. 1d, near-zero MV). The price-to-walk curves for the walking-no-exo and exo-powered conditions for all participants are shown in Supplementary Fig. 2; the price-to-walk curves for the exo-powered-off condition for the corresponding participants are shown in Supplementary Fig. 3.

Our approach was able to quantify the intuitive effects of added mass and assistance. Across subjects, the MV of the exoskeleton + assistance was positive but not significant, while the MV of the unpowered exoskeleton was significantly negative, and the MV of the assistance itself was significantly positive. The average inter-subject MV of exoskeleton use was 5.8%, with a standard deviation (SD) of 31.14% ($N = 16$, SEM = 7.8%, Fig. 2a). The exoskeleton + assistance thus provided only a small value benefit to the average participant. Using a two-tailed t-test, the average MV of exoskeleton + assistance (5.8%) was not significantly different from zero ($p = 0.24$, effect size: 0.18). However, as denoted by the high standard deviation, some participants received large benefits from the device's assistance, while others experienced an economic penalty from exoskeleton use. The average MV for the unpowered exoskeleton was −31.8% with a standard deviation of 45.0% ($N = 10$, SEM = 14.2%, Fig. 2b). Using the same two-tailed t-test as in the exo-powered condition, we found this change to be significantly different from zero ($p = 0.03$, effect size: 0.67). In addition, the powered assistance alone from the exoskeleton provided a significant increase in value (mean: 33.8%, SD: 38.1%, SEM = 12.0%, $p = 0.01$, effect size: 0.84, $N = 10$, Fig. 2c).

The integral of each price to walk curve yields the cumulative price to walk for each condition (Supplementary Fig. 4). The

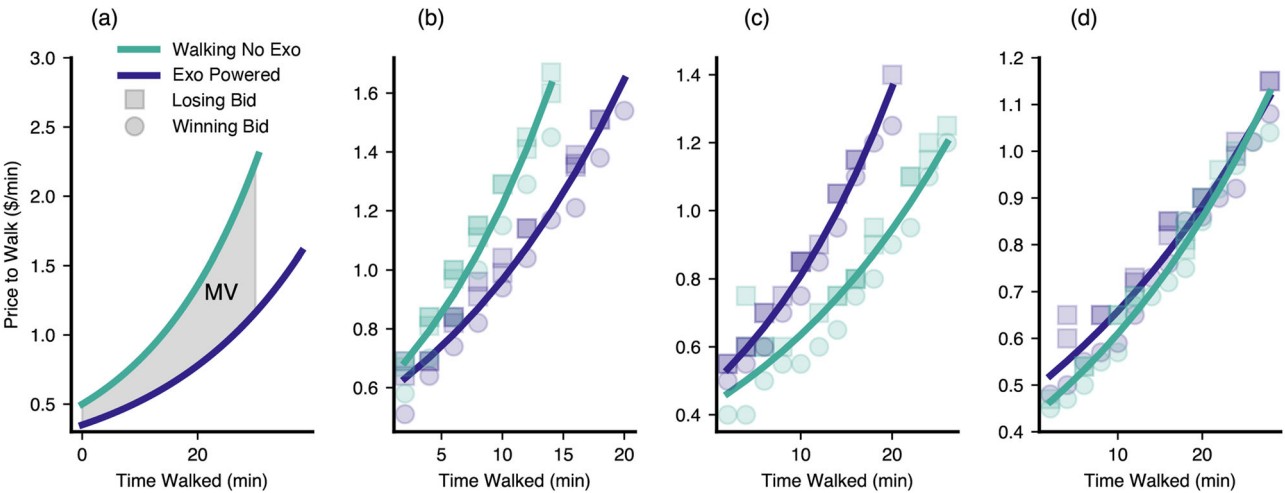

**Fig. 1 Specific examples of price to walk curves for different conditions. a** Representative price to walk curves for the walking-no-exo condition (teal) and the exo-powered condition (purple) from different subjects. The marginal value (MV, the value of the exoskeleton assistance) is given by the area between the curves. In actual trials, the price to walk curves are estimated by fitting first-order exponentials to subject bids. Circles denote winning bids, while squares denote losing bids. The participant in (**b**) shows a clear benefit from the device, the one in (**c**) shows a clear detriment, and the one in (**d**) is more ambiguous.

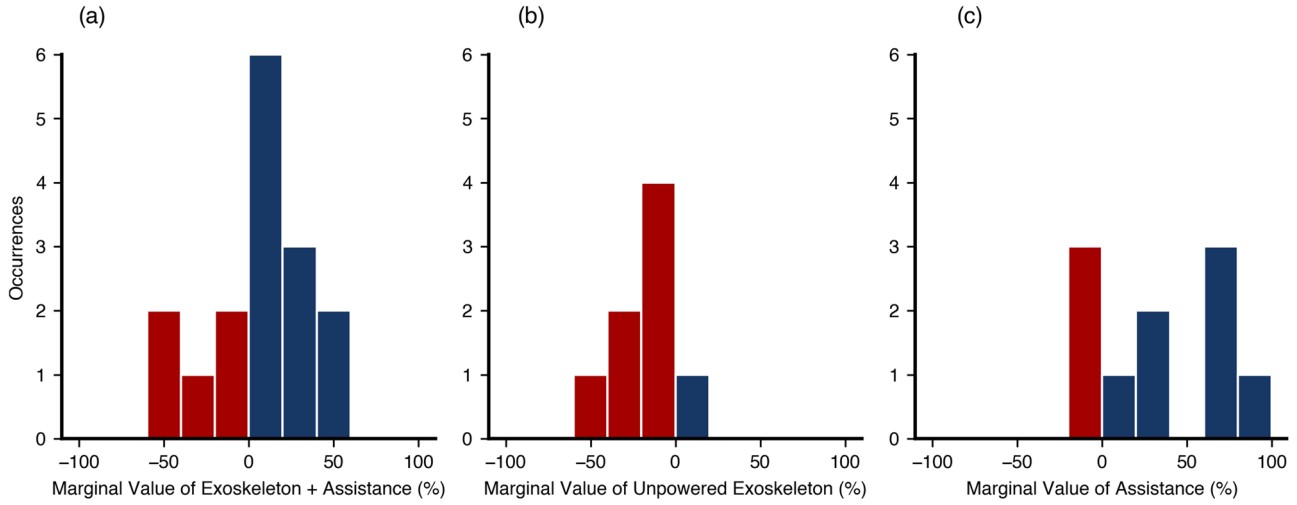

**Fig. 2 Histograms of the Marginal Values (MVs) for the different conditions.** Positive MVs, indicating that value was added to the wearer are in blue, while negative MVs, indicating that costs were imposed on the wearer, are in red. **a** The MVs of exoskeleton + assistance for all sixteen subjects. The average MV was 5.81%, with an SD of 31.1%. **b** The MVs of the unpowered exoskeleton for the twelve subjects who participated in this condition. Aside from one subject, all participants experienced notable economic costs from this condition, reflected by the average MV being significantly negative (average: −31.8%, SD: 45.0%, $N = 10$ participants). **c** The MVs for the exoskeleton assistance alone (average: 33.8%, SD: 38.1% $N = 10$ participants). While the average MV of the exoskeleton + assistance was not significantly positive, the assistance alone conferred a significant benefit.

average cumulative price for the walking-no-exo condition was $29.20 ± $28.10 (Supplementary Fig. 4a, Supplementary Table 1), for the exo-powered condition $25.40 ± $13.90 (Supplementary Fig. 4b, Supplementary Table 1), and for the exo-powered-off $49.70 ± $54.30 (Supplementary Fig. 4c, Supplementary Table 2). Changes in participant value due to the different walking conditions can be measured by comparing the associated cumulative prices.

We also characterized the repeatability of our measurements with a subset of participants to verify that the changes in price to walk detected due to the changing walking conditions were attributable to the difference in perceived difficulty between conditions and not due to participant day-to-day variability (a representative repeater trial is shown in Fig. 3, all trials shown in Supplementary Fig. 5 and given in Supplementary Table 3). For the four subjects that repeated the walking-no-exo condition at

least once, the average intra-subject standard deviation of the cumulative price for the walking-no-exo condition was 3.38%, expressed as a percentage of each subject's average walking-no-exo cumulative price. This quantity represents the day-to-day fluctuation in the value demanded to walk, in addition to any user or experimentally derived noise.

## Discussion

In this work, we use the Vickrey second-price auction as a method to capture the economic value or detriment provided by exoskeletons and their assistance in the immediate term. As part of our broader goal of emphasizing the user's role in the design, control, and evaluation of exoskeletons[35,56], the intent of this work is to quantify the success of these technologies through a user-centered metric which encompasses the different aspects of

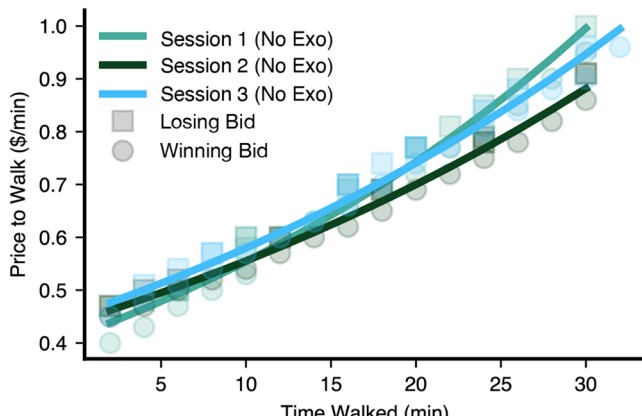

**Fig. 3 Representative price-to-walk curves for a single 'repeater' subject across different days.** The agreement across the curves demonstrates the test re-test reliability of our approach. These sessions were completed over a 7-day span.

the user experience, including exertion, weight, comfort, and assistance. To this end, the wearer specified their "price to walk" during a series of Vickrey auctions from which we quantified the economic value of different conditions. Our approach is particularly relevant for assessing the success of augmentative exoskeletons, which do not have a clear, clinically-relevant, biomechanical or physiological objective (as in the case of prostheses or orthoses). Our work is motivated by the belief that obtaining accessible and relevant metrics of success is critical to the successful development and adoption of exoskeletons in the real world. This motivation is supported by a history of strong interconnection between the assessment of exoskeletons and their design and control architectures[24–30]. Our strategy quantifies success using monetary currency, as opposed to biomechanical or physiological quantities that may be more difficult for users, manufacturers, and those outside of the exoskeleton research field to interpret.

Our results showed that the economic value provided by the exoskeleton assistance was comparable to the cost incurred by wearing the unpowered system. We obtained these results by quantifying the marginal value (MV) between the different conditions, which captured the economic benefit or detriment between these conditions. To isolate the net value of the exoskeleton + assistance, we directly calculated the difference in value between unassisted walking, which acts as a baseline control condition, and exoskeleton-assisted walking, which introduces the economic value provided by the exoskeleton. To calculate the cost of wearing the unpowered exoskeleton, we compared the values of unassisted walking to that of walking with the unpowered device. The value of the assistance alone was then isolated by comparing the price-to-walk curves of the exoskeleton-assisted condition and the unpowered condition, which controlled for the value of the participants' time and thus enabled assessment of the detriment caused by the device's mass. The value of each condition in dollars/hr was obtained by expressing the average cumulative price as a rate of dollars/hr for the unassisted condition, and multiplying this rate by the MV of each condition, expressed in percentage change from the unassisted condition. The resulting hourly rates therefore represent the change in dollar value from the baseline unassisted condition. While every subject completed at least the unassisted condition and the powered exoskeleton condition, not every participant completed the unpowered condition, and thus we cannot simply take the difference between the average value of the assistance and the average penalty of the exoskeleton's mass to obtain the net value.

The average cumulative price of walking uphill for 30 min without the exoskeleton was \$29.20 (\$58.50/h SD: \$57.50/h). The MV of wearing the unpowered exoskeleton—that was not providing assistance—was −31.8%, which translates to a monetary cost of \$18.60/h (SD: \$18.30/h) for wearing the unpowered system; this value in dollars was calculated by multiplying the average cumulative price per hour (\$58.50/h) and the MV of the unpowered exoskeleton (−31.8%, $N = 10$). When assistance was applied by the exoskeleton, the MV increased to just above zero (\$3.40/h, SD:\$3.40/h) when compared to not wearing the exoskeleton, which was calculated by multiplying the average cumulative price by the MV of exoskeleton use (5.8%, $N = 16$). The marginal value added by the assistance alone was 33.8%, which translates to an added value of \$19.80/h (SD: \$19.40/h); which was calculated by multiplying the average cumulative price by the MV for the assistance alone (33.8%, $N = 10$). Thus, the assistance applied by the exoskeleton offset the cost of wearing the system, and the net benefit was modest but positive. These results suggest that modern augmentative ankle exoskeletons may not provide a substantial benefit to their wearers during short-term uphill walking. This is particularly surprising because of the lightweight, refined design of the exoskeletons used, and the high physiological demands of the uphill walking task[57,58], which we chose to increase the observed value of the exoskeleton (i.e., to improve the signal-to-noise ratio in our measurement of economic value).

Exoskeleton controllers can potentially be directly engineered to maximize the MV added (in dollars) from the assistance of the exoskeleton during walking. Similarly, exoskeletons can themselves be designed to minimize the economic cost they impose on the wearer, which includes not only weight, but also other factors such as inertia and discomfort. This information can be obtained using the MV of the unpowered exoskeleton when compared to the walking-no-exo condition. In addition, this study suggests that exoskeleton development may be improved by identifying the users who innately receive greater economic value from wearing the system. We found that nine participants obtained substantial positive economic value from wearing the system (27.80% ± 16.30%, corresponding to \$16.30/h ± \$9.50/h), even though the average across all participants was lower. Such "responders" may be more willing to adopt exoskeletons in the face of potential drawbacks, including cost and added mass, among others. Understanding why some users obtain this value would enable the targeting of these individuals to maximize the translation and impact of augmentative exoskeletons. For example, models such as the diffusion of innovation (DOI) theory within the social sciences categorize individuals based on their willingness to adopt a new innovation[40], such as exoskeletons. It is possible that the individuals who obtained positive value within our study fall into the "early adopter" category, and are more willing to find value in exoskeleton use. Future investigations could focus on the separation of these individuals a priori based on the social characteristics within DOI theory, or based on other human factors, such as fitness level or perceptual abilities[35]. Furthermore, investigations of whether users can be "trained" to receive a greater economic value from the exoskeleton are promising avenues of future study. With respect to the current standard of metabolic cost reductions, the metric of MV and the notion of the price-to-walk more holistically quantify the experience of exoskeleton use, including subjective evaluations, to a greater degree than singular physiological objectives. Thus, our work motivates future investigations to discover the degree to which economic value encodes metabolic benefits of exoskeletons.

Participant results were consistent when repeated across days, supporting the quality of our measurements. Four participants repeated the walking-no-exo condition across several days, and the inter-day variance of the cumulative prices was low (\$0.67,

Supplementary Table 3). This consistency supports the ability for our approach to quantify perceived changes in value, with potentially minimal effect from inter-day confounding factors. Using this inter-day variance, we estimated the Minimum Detectable Change (MDC), defined as the minimum change in value not caused by chance. The MDC for the MV measurement was 9.4% (95% confidence interval, difference between two measurements, standard deviation 3.4%). To identify which subjects had a noticeable change in value, we compared each participant's MV against the inter-day variability's 95% confidence interval (±9.4%); nine of 16 subjects had MVs exceeding this threshold in the positive direction, while four subjects exceeded it in the negative direction.

Our auction strategy avoids several potential limitations of more direct methods to quantify value, such as a Likert scale test, direct feedback, or auctions without a real monetary payout[59]. The locomotion task implemented in our approach does not necessitate extreme fatigue and can remain relatively short, avoiding confounding factors (e.g., boredom or opportunity cost). In addition, the incentive structure of the auction links participant bids to a specific consequence (e.g., walking uphill), and are thus less prone to biases, such as self-enhancement or social desirability, while still remaining intuitive[53,60].

Individual bids and, subsequently, cumulative prices varied widely across subjects. Since participants were able to set their own bids, this added inherent variability across subjects. This variation may have resulted from each participant having different internal valuations of their time, which could have been driven by differing socioeconomic status, athleticism, opportunity cost, or other factors. This large discrepancy motivated the creation of the marginal value (MV) metric, which normalizes by each participant's 'baseline' cumulative price from the walking-no-exo condition, expressing the change as a percent of the walking-no-exo condition.

We anticipate that the conclusions drawn in this study extend to other augmentative ankle exoskeletons, although further study is needed to quantify the value from other technologies. The ankle exoskeletons from this work were developed for commercial use, and represent a "best-in-class" technology; the system's refined design is lightweight, untethered, and provides substantial net-positive energy during each stride (13.4 ± 2.9 J). We expect that other, similar exoskeletons would have comparable marginal values, and thus may also not provide a substantial economic benefit to their wearers. In addition, although the task of uphill walking does not represent all possible uses for augmentative exoskeletons, it enables an opportunity to quantify value provided by the exoskeleton during an intuitive application where it can provide substantial benefit. We expected that the increased energetic difficulty of uphill walking[57,58] would enable the exoskeleton to more readily demonstrate its value to the wearer. Furthermore, we chose the uphill walking task to reduce experimental duration as the greater difficulty would cause participant bids to rise more quickly. We expect that for less strenuous tasks, exoskeletons will show reduced marginal value; this hypothesis is supported by the results of a separate study we conducted (see Supplementary Methods) in which the MVs of the exoskeleton + assistance were lower for level-ground walking when compared to uphill walking. Future studies could employ the Vickrey auction protocol to quantify the value of various different exoskeletons during different, more realistic use cases to gain a better understanding of when and which exoskeletons are perceived as valuable by users.

The changes in exoskeleton conditions caused intuitive changes in value, supporting the validity of our economic value metric. That is, we believe the different exoskeleton conditions (walking-no-exo, exo-powered-off, and exo-powered) likely drove the changes in value measured, and these changes agree with the biomechanical demands of the conditions. The MV of the exo-powered-off condition was strongly negative, denoting a cost to wearing the system. This result is expected, since without assistance, wearing an exoskeleton is akin to wearing shank weights during locomotion, which would necessitate greater mechanical work from the triceps surae and cause an upstream increase in (by ~16.8 W based on the results from refs. [38,61]). In addition, when the assistance was added, the value increased to just above zero, indicating the assistance was useful in offsetting the challenge of wearing the unpowered system during uphill walking.

The MV for the exo-powered condition may shift with repeated sessions. This could be caused by several factors; for example, participants may adapt to the exoskeleton assistance if given more time to walk with the device. Prior work has found that training sessions across multiple days yield greater reductions in metabolic rate in naive users[2]. Accordingly, the MV for the exoskeleton may adjust as well while the user adapts to the experience of wearing an exoskeleton. Similarly, it is possible that the initial MV may be have been partly driven by the positive value a user places on the novelty of wearing an exoskeleton, if the experience was novel for them. In our protocol, to reduce the influence of this possibility, we allowed the participants to complete an adaptation period in which they freely walked with the exoskeleton prior to undergoing the full experiment. For this initial investigation, we sought to understand the immediate economic value obtained from a first-time experience of exoskeleton use, analogous to a user assessing an exoskeleton when making the decision to adopt (i.e., a "test drive"). Future work is needed to understand any adaptation of value that may occur over time, which would have implications in the longer-term value of these technologies.

The use of simulated bidding agents may have influenced the results but we expect the overarching conclusions of the study would not be affected. In our auction protocol, we used simulated bidding agents ("robo-bidders") to model human behavior, rather than implementing our approach with multiple human subjects. The intent for this choice was to reduce the logistical challenges of our approach, which would otherwise have required multiple treadmills and exoskeletons. To mitigate any effect of the simulated nature of the other participants, the human participant was informed that the robo-bidders were humans participating in the experiment in remote locations. The behavior of the robo-bidders —defined by a parameter that governs the rise and fall of their bids—was derived from pilot data obtained from human participants. To reduce the likelihood that the human participants could infer the robo-bidder behavior model, the robo-bidder bids were corrupted with noise. The use of robo-bidders also enabled us to standardize the interaction between the human participant and the other auction participants. Any series of auctions would naturally establish an equilibrium between the participants; thus, by using robo-bidders, we were able to control for this equilibrium, which strengthens our ability to compare across subjects. Although we expect that, given honest bidding, participant values will remain constant, future work should investigate the effect of changing robo-bidder parameters on participant bids. In addition, the use of robo-bidders enables our results to more readily be compared across other researchers, institutions, and exoskeletons that are assessed using comparable methods. The code to implement the robo-bidders is provided in ref. [62].

During the experimental protocol, participants responded with bids as instructed, but we are unable to know for certain that their bids were truly honest (i.e., truthfully reflecting their internal sense of value). Our study relies on participants honestly reporting their bids, which is theoretically guaranteed as an optimal strategy by the nature of the Vickrey second-price

auction for rational actors[46]. In addition to the natural structure of the Vickrey auction, which incentivizes truthful bidding, we minimized the potential influence of dishonest bidding by setting the auction interval to 2 min (as opposed to a shorter duration). We chose the 2-min duration to increase the effort required, and thus minimize potentially 'dishonest' exploratory bidding, which would corrupt the price to walk curves. In addition, subjects continuously received verbal instructions to always bid honestly as the best strategy, which has been demonstrated to increase the likelihood of honest bidding[63]. Participants received the expected monetary compensation that resulted from their winning bids, which similarly incentivized truthfulness. The presence of automated robo-bidders, rather than actual humans, rendered our protocol similar to an auction that used the Becker-DeGroot-Marschak (BDM) method[64], in which participants compete in auctions against bids generated randomly via a statistical distribution, rather than other humans. It has been shown that the BDM auction is incentive compatible with the strategy of truthful bidding to sell an item (as in the Vickrey auction) when the maximum buyout price—the maximum the seller could be expected to receive—generated by the distribution does not exceed a realistic buyout price for the good and when the sellers are aware of this concept[65]. However, we note that unlike in typical BDM auctions, our approach features repeated Vickrey auctions with realistic competing bids, rather than stochastically-generated bids. Thus, after each auction when the winning bid was disclosed, subjects had a general understanding of the range of bids that would be expected in the experiment. In addition, even if the participant bids given were greater than honest bids, the net MVs calculated would be similar due to the MV metric being calculated using the difference of cumulative prices (provided the magnitude of the effect did not vary across days). A future study that replicates the protocols in this work while including all human participants would be informative.

## Conclusion

We have developed a method to quantify the economic value of augmentative exoskeletons, and used these methods to assess the value provided by bilateral ankle exoskeletons during uphill walking. Our results underscore the challenge of developing exoskeletons that provide a clear, meaningful benefit when augmenting the healthy human neuromotor system. The value of the assistance provided by the exoskeletons was modest, and just offset the cost of wearing the unpowered system. Our results also suggest that more work is needed to identify why some participants received substantially greater value from the exoskeleton assistance, which could be used to identify or train individuals for maximizing the real-world impact of these technologies. Finally, the economic value metric we have developed can be readily used to compare different design and control strategies to develop exoskeletons that are maximally valuable to their wearers.

## Methods

**Participants**. In this study, 16 able-bodied participants ($N = 16$, 4 female, 12 male; age = 26.3 ± 4.6 years; mass = 77.1 ± 13.4 kg, Table 1) walked using bilateral ankle exoskeletons on a treadmill. All participants provided written informed consent before participation. The study protocol was approved and overseen by the Institutional Review Board of the University of Michigan (Study ID: HUM00158854). Participants had no prior experience walking with the bilateral ankle exoskeletons featured in this study.

**Exoskeleton apparatus**. Our approach quantifies the value of bilateral ankle exoskeletons that were designed to improve the energetic efficiency of human walking (ExoBoot, Dephy Inc., Maynard, MA, Fig. 4). The commercially-available system utilizes an onboard brushless electric motor and flat cable transmission (for a mean transmission ratio ~15:1, Fig. 5) to apply ankle assistance during walking. The exoskeletons have a single-powered degree of freedom (dorsi-plantar flexion) and a passive, unactuated degree of freedom (inversion-eversion). The

| Table 1 Participant characteristics. | | | | |
|---|---|---|---|---|
| **Participant** | **Gender** | **Weight (kg)** | **Age** | **Height (cm)** |
| 1 | F | 83.6 | 24 | 160.0 |
| 2 | M | 99.2 | 27 | 176.0 |
| 3 | M | 84 | 22 | 186.0 |
| 4 | M | 90.7 | 25 | 182.9 |
| 5 | M | 80.7 | 26 | 177.8 |
| 6 | F | 72.6 | 29 | 170.2 |
| 7 | M | 80 | 34 | 180.3 |
| 8 | M | 74.8 | 34 | 177.8 |
| 9 | M | 68.0 | 25 | 170.2 |
| 10 | M | 70.3 | 22 | 185.4 |
| 11 | M | 67.1 | 25 | 167.6 |
| 12 | M | 104.3 | 36 | 193.0 |
| 13 | F | 74.8 | 23 | 177.8 |
| 14 | M | 70.3 | 21 | 172.7 |
| 15 | M | 61.2 | 23 | 163.0 |
| 16 | F | 52.4 | 25 | 162.6 |

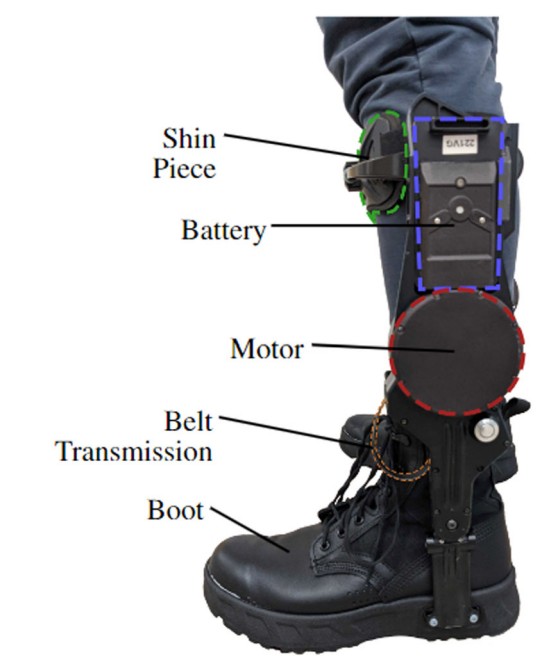

**Fig. 4 The Dephy ExoBoot used in this experiment.** An electric motor applies plantarflexion torques at the ankle via a belt transmission.

transmission is unidirectional, which enables the system to apply plantar flexion assistance torque but it cannot provide dorsiflexion assistance. Each side of the exoskeleton applies a torque profile (Fig. 6a) that provides a burst of positive power (Fig. 6b) during the terminal stance phase of gait, augmenting the propulsive effort provided by the triceps surae. The average energy provided by the exoskeleton during the gait cycle is 13.4 ± 2.9 J. During swing phase, the exoskeleton is able to add slack to the belt drive, thereby preventing any unwanted resistance to the foot; this capability stems from the unidirectional nature of the design. Gait progression, inferred from heel-strike timing events, was used to schedule how assistance was provided during each step. Similar exoskeletons have been shown to lower the user's metabolic expenditure during walking as well as reduce the biomechanical power requirements at the ankle and other joints of the legs[1,32,38]. The walking assistance controller was developed by Dephy Inc. as part of their commercial system. As one of the first commercially available exoskeletons, the Dephy Exoboot was chosen for its ease of use, robustness, and representation of the state-of-the-art.

**Walking protocol**. Participants received guidance on the Vickrey auction protocol prior to the experiment. During this time, participants read a lay explanation of the Vickrey auction. Subsequently, participants underwent two mock Vickrey auctions, where the optimal strategy of truthful bidding was explained. Subjects were also repeatedly informed of the optimality of the honest-bid strategy to improve the

number of honest bids provided[63]. In the first mock auction, participants were presented with a miscellaneous office supply item and told to bid honestly on it as if they were competing to purchase it in a real Vickrey auction. Each participant wrote down their bid on an index card, while the researcher wrote down some artificial competing bids. The participant then revealed their bid, and the experimenter explained which bid won (the highest bid) and what the cost would be (the second-highest bid). Next, participants were told to imagine they were participating in the actual study, in which they would compete to sell their walking time in exchange for monetary compensation. Participants were prompted to consider

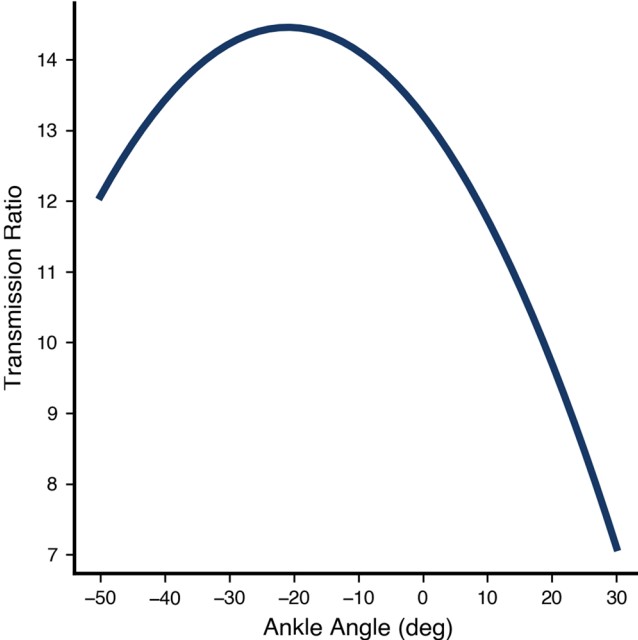

**Fig. 5 The transmission ratio curve of the exoskeleton device used in this experiment.** Positive ankle angles denote dorsiflexion and negative angles denote plantarflexion. The curve was modeled as a second-order polynomial, and was obtained by taking the derivative of the best-fit third-order polynomial that relates motor angle to ankle angle.

bidding based on an hourly wage, although they were also told that any truthful bid would be accepted. Again, participants wrote down their bids while the researcher aggregated the competing bids. As in the previous mock auction, the participants revealed their bids, and the experimenter walked them through which bid won and the payout.

Following familiarization with the exoskeleton and auction mechanics, participants competed in sequential Vickrey auctions to sell their time while walking uphill (Fig. 7). The protocol was organized into a set of Vickrey auctions experienced in series, each lasting 2 min. If the participant won the auction, they would walk for the 2-min interval, and they would rest if they lost the auction. During walking, participants walked on a split-belt treadmill with a 10° incline. Unlike in the mock auctions, the participants would accrue and receive monetary compensation according to the Vickrey auction protocol. At the end of each interval, the winning bid (second-lowest bid) was revealed; while in a traditional Vickrey second-price auction, all bids would be revealed, the participant was only given the winning bid to avoid unduly influencing them toward a specific bid range or incentivizing the participant to guess a pattern of the competing bids. The sequential auctions lasted a randomly-specified duration between 50 and 70 min, with a mean duration of 60 min; participants walked on average a total of 31.3 min ± 11.5 min. The duration uncertainty was added to discourage subjects from waiting until their bidding competitors were "exhausted" to inflate payouts. The 2-min interval duration was chosen to minimize subject exploratory behavior. In total, participants experienced ~25–35 Vickrey auctions in series. By aggregating each subject's bids across the series auctions, the resulting price to walk curve captured the participants' valuation of their time to complete the experiment, which can be used to provide insight into the value provided by the exoskeleton in this application.

Our protocol was broken up across several days to reduce the effect of fatigue from the experiments. The order of the testing was randomized across participants. On one day, subjects completed the Vickrey auction protocol with normal shoes (i.e., the walking-no-exo condition); these data establish their baseline price to walk curve and cumulative price. On a different day, subjects instead walked with powered exoskeleton assistance (i.e., the exo-powered condition). Participants were not explicitly told that the exoskeleton would provide assistance; instead, they were told that the exoskeleton was applying a randomized torque profile that would either help or hinder their locomotion. This was done to reduce the chance that participants would experience a placebo effect and perceive the exoskeleton as valuable due to information provided by the experimenter. By comparing the price to walk curves from the walking-no-exo and exo-powered conditions, we obtained a measurement for the value provided by the exoskeleton in this task. If undergoing the exo-powered condition on the second day, subjects were given time to re-familiarize and experience the exoskeleton's assistance.

In addition, each subject was randomly assigned to additional experimental sessions to investigate other attributes our approach. Participants were randomly assigned to groups that either quantified the inter-day variability of the price-to-walk measurements, or investigated the additional condition of walking with the exoskeleton without assistance (i.e., the exo-powered-off condition). Due to the

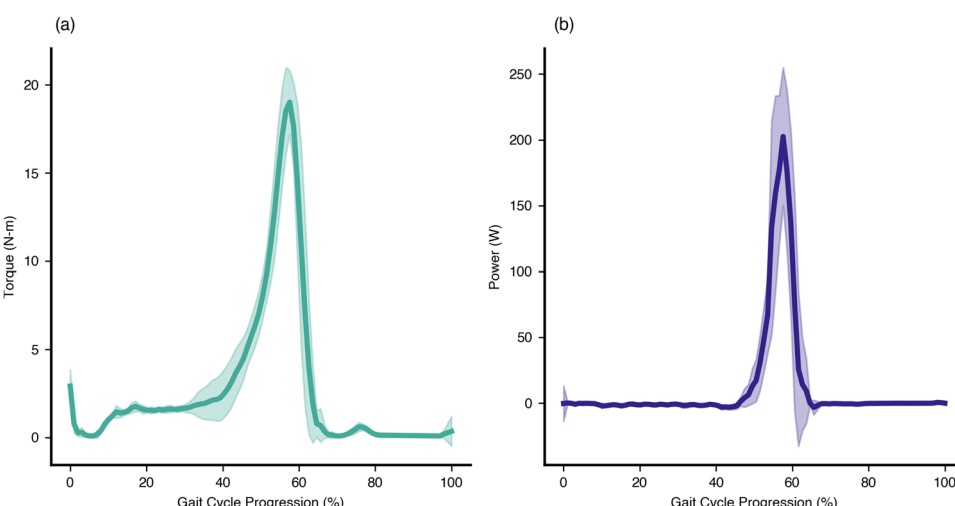

**Fig. 6 The torque and power assistance profiles applied by the exoskeleton used in this study. a** The torque profile applied by the bilateral ankle exoskeletons during the uphill walking task. The torque applied stems from the proprietary control strategy implemented by Dephy Inc and represents the state of the art. The mean profile is shown in solid teal, with a single standard deviation shown by the shaded region. Torque was quantified by using the measured q-axis motor current and the experimentally-derived, q-axis torque constant to calculate motor torque[66], and then multiplying that value by the instantaneous transmission ratio. Gait cycle progression is defined as beginning and ending at sequential ipsilateral heel-strikes. **b** The power applied by the exoskeletons at the ankle during the walking trial. Power was calculated by multiplying the current-derived torque profiles in (**a**) by measured ankle angular velocities. The mean profile is shown in solid purple, with a single standard deviation shown by the shaded region. The average energy provided by the exoskeleton, obtained by integrating the power curves over time is 13.4 ± 2.9 J.

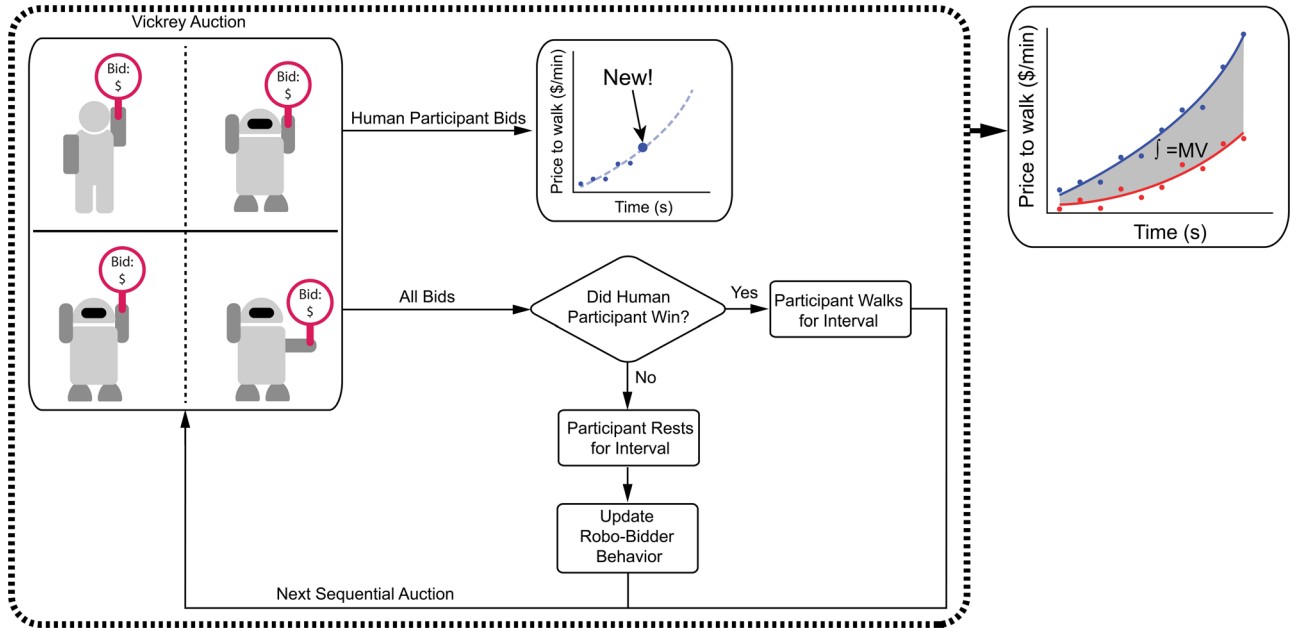

**Fig. 7 A diagram of the experimental Vickrey auction protocol.** In this protocol (black dashed block), human participants compete against computerized bidders in sequential Vickrey auctions for monetary compensation in exchange for undergoing a strenuous locomotion task. A human participant, shown in the upper-left quadrant of the 2 × 2 grid, places bids to compete against computerized (`robo-') bidders, represented by the robots in the remaining three quadrants. Should the human win, they accrue the payout and walk for the 2-min interval; should they lose, they instead miss out on the payout and rest until the next interval. Regardless of whether the human won or suffered a loss, their bid (solid blue circle) is aggregated to form their price to walk curve. As the human fatigues, their price to walk to increases (blue dashed line). Participants competed while wearing normal shoes (`no-exo' condition, blue solid line in the solid black block) and while receiving powered assistance from the exoskeleton (`exo-powered', red solid line in the solid black block). The integral of the difference between the price-to-walk curves for the no-exo and exo-powered conditions is the Marginal Value (MV) added from the device's assistance (shaded region between solid rend and blue lines in the solid black block).

number of experimental sessions required, participants were assigned to only one of these groups. Of the sixteen total participants, ten participants walked in the exo-powered-off condition, while four participants repeated the walking-no-exo condition at least once (the remaining two subjects were not able to participate in the additional sessions). Within the experimental group that repeated the walking-no-exo condition, three participants repeated the condition twice more, while the last participant repeated the condition once. The intent of investigating the exo-powered-off condition was to support that the price-to-walk curves captured the cost of wearing the additional exoskeleton weight, when no assistance was provided. The goal of repeating the walking-no-exo condition was to provide insight into the test re-test variability of the price to walk curves and how they may be affected by inter-day confounding factors.

**Robo-bidders**. For a live Vickrey auction, multiple participants are needed; however, this constraint adds practical challenges in coordination and logistics. To this end, we utilized computerized bidding agents (robo-bidders) as a substitute for other human participants in the Vickrey auctions. The mechanics of the robo-bidders are modeled on the bidding behavior of three pilot subjects, which featured an expected trend of increased bids as they walked for longer on the uphill walking task. The robo-bidders bid with prices following a first-order exponential function of the number of intervals spent walking. This model captured the effect of fatigue and increased their bids as the robo-bidder won the auction. If the robo-bidder did not win the auction (meaning the human participant won), their bid remained constant. Gaussian white noise (zero mean, standard deviation $0.01) was added to corrupt the robo-bidder bids, to reduce the likelihood that the human participant would intuit the robo-bidder model. Robo-bidder behavior at time $t_k$ during simulated walking was governed by the following equation:

$$y_k(t) = k \cdot e^{(b \cdot t_k)} + \sigma, \qquad (1)$$

where $y_k$ was the robo-bidder's price, $k$ was the initial price, $b$ was the rate at which the price increases, is time, and $\sigma$ was drawn from normal distribution $\mathcal{N}(0, 0.01)$. Parameters $k$ and $b$ were set differently for each robo-bidder; full implementation details can be found in [TBD]. Having robo-bidders instead of human auction participants reduced the logistical difficulty of executing the Vickrey auction experiments over time, while still replicating modeled behavior of a human participant. Naturally, the robo-bidder bids competed in parallel with the human subject and approached an equilibrium behavior, which affected the total walking time of the human participant. In the application of auctions in series, an equilibrium would also likely be established between multiple human participants.

The human participants were not initially made aware of the fact that they were competing against computerized opponents. They instead were told that they were competing in live Vickrey auctions against other humans located remotely, with the experimenters communicating live. This was done to prevent the participants from attempting to learn and exploit the price to walk curves of the robo-bidders in an attempt to maximize profits beyond the strategy of honest bidding. Participants were debriefed on the true nature of the robo-bidders at the conclusion of their participation in the experiment.

**Marginal value**. Our outcome metric is the marginal value (MV) that stems from difference in two price to walk curves obtained for different experimental conditions. Each price to walk curve was fit with a first-order exponential response of the form:

$$Y(t) = k \cdot e^{(b \cdot c \cdot t)}, \qquad (2)$$

where $Y$ is the participant's price, $k$ is the initial price, $b$ is the rate at which the price increases, $c$ is a scaling factor equal to the participant's win-rate, and $t$ is time. The scaling factor is necessary to control for the different durations of walking experienced by the participants (i.e., variations in the number of auctions won). After correcting for this win-rate, these curves are integrated and subtracted, yielding the area between the two curves. This area represents the marginal value added—in US dollars—from one exoskeleton condition compared to another. The expression for the MV is as follows:

$$MV = \overbrace{\int_{t_1}^{t_2} k_1 \cdot e^{(b_1 \cdot t)} dt}^{\text{exo condition 1}} - \overbrace{\int_{t_1}^{t_2} k_2 \cdot e^{(b_2 \cdot t)} dt}^{\text{exo condition 2}}, \qquad (3)$$

where $k_1, b_1$ correspond to the first exoskeleton condition in the comparison, $k_2, b_2$ correspond to the second condition, and $t_1, t_2$ are the bounds of the time domain for the integrals—0 and 30 min, respectively, which roughly corresponds to the average time each participant walked in each trial. The MV is then equivalent to the value added or removed by the exoskeleton during a continuous 30-min, uphill walking task. We commonly normalized the MV by the cumulative price from the walking-no-exo condition to obtain a percentage change. However, this metric can be converted to compare any two experimental conditions or other candidate control strategies that researchers wish to evaluate in terms of economic value.

**Reporting summary**. Further information on research design is available in the Nature Portfolio Reporting Summary linked to this article.

## Data availability

All data needed to support and evaluate the conclusions of the paper are available in the paper, the Supplementary Materials, our CodeOcean repository (https://codeocean.com/capsule/7159524/tree), our Zenodo repository (https://doi.org/10.5281/zenodo.7922805) or from the corresponding author upon reasonable request.

## Code availability

The code necessary to implement the robo-bidders used to simulate the Vickrey Auction in this paper is provided in our CodeOcean repository (https://codeocean.com/capsule/7159524/tree) and in our Zenodo repository (https://doi.org/10.5281/zenodo.7922805).

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

## Author contributions

R.L.M., G.C.T., D.M., and E.J.R. designed the study; R.L.M. performed the experiment and analyzed data with input from G.C.T., D.M., and E.J.R.; R.L.M., G.C.T., and E.J.R. wrote the paper. E.J.R. directed the study and all authors approved the final version of the manuscript.

## Competing interests

The authors declare no competing interests.

## Ethics approval and consent to participate

All experiments were carried out with informed consent at the University of Michigan, with approval from the Institutional Review Board of the University of Michigan Medical School (IRBMED).
