## [Peer Review File · Communications Engineering]

The Economic Value of Augmentative Exoskeletons and their AssistanceReviewers' comments:

Reviewer #1 (Remarks to the Author):

Review for Communications Engineering of “The Economic Value of Augmentative Exoskeletons and their Assistance,” a paper written by Roberto Leo Medrano, Gray Cortright Thomas, Drew Margolin, and Elliott J. Rouse (MTMR)

In second-price auctions (SPA), bidders have dominant strategies to reveal their valuations for the object auctioned off. If one is willing to assume that bidders rely on such strategies, bids will reveal valuations. For example, if a car is auctioned off under SPA rules, then submitted bids will reveal bidders' valuations. MTMR argue, compellingly in my view, that it may be useful to figure out how much value users of exoskeletons attach to such devices. Moreover, SPA may provide useful measuring rods in this connection. If it is the case the engineers have never thoughts along these lines before, then this message seems pioneering and worth having. I then support publication of a version of the paper. However, the manuscript, as currently formulated, has some serious flaws which need to be corrected. My comments below are meant to help MTMR fix the problems:

1. SPA are useful as valuation measuring rods only if it is interesting to learn the value of the item auctioned off. That might clearly be the case in my above example with a car, say if one is a company like Toyota and interested in figuring out demand for one's vehicles. It is furthermore compelling, as I said above, that someone may want to know the value that users of exoskeletons attach to such devices, in certain situations. However, I very much doubt that the situations studied in MTMR's experiment belongs to that class. Subjects buy (or sell) the right (or obligation) to take a “walk” on a treadmill wearing an exoskeleton. That seems to be a rather uninteresting or irrelevant item to buy or sell. For example, in the movie *Edge of Tomorrow*, Tom Cruise's character uses exoskeletons to fight off alien “Mimics,” and it is clear that this is quite valuable to him. However, I doubt that he would have wanted to walk on treadmills instead. Accordingly, I feel that this study does not really produce any insights as regards the actual value of the use of exoskeletons in meaningful situations. In there is a contribution, it will rather concern some kind of proof-of-concept. The study alerts readers to the possibility of using SPA for evaluating engineered items, but there is no insight beyond that. The manuscript needs to be re-written to reflect this outlook.

2. Presumably, most readers of this journal do not know about SPA and their relevance for measuring valuations. A key objective of this paper should therefore be to clearly explain the method. However, the presentation on p4 is not very clear, including as regards conveying/proving the central insight that bidders have dominant strategies to equate their bids to their values. The presentation on p14 is better, but too relegated. Having both presentations is unnecessary and repetitive.

3. On p4 MTMR explain how SPA is a “sealed-bid” auction, which is true. However, MTMR get it wrong what is the defining characteristic of that property. Namely, it is NOT that “bids are not revealed publicly.” Rather, it is that no bids are revealed to any bidder before that bidder submits his own bid. That is, others bids are sealed to a bidder when he submit his own bid. MTMR, by contrast, write as if it is critical that bidders are not told each others’ bids after the auction is over, which is not the case.

4. MTMR’s exercise can be understood only if the SPA that is used is actually described. One key aspect is the item being auctioned off. What is it? Most awkwardly, MTMR never explain this in the main text. One has to flip back to p12+ to learn about this. The paper cannot be understood unless the auction details are described, and, of course, this material has to be presented in the main text for the paper to be readable.

5. As I read, for a long time it wasn’t clear to me that MTMR ran any auction at all, in the sense that if someone did not win the auction then they didn’t get the item being auctioned off. Without that key feature, the whole study would have been flawed, and it would have been necessary to reject the submission. Now, MTMR did not make such a mistake. However, this is not revealed until p16, in the caption under Figure 7 (“should they lose, they instead...”). Overall, the presentation is obscure as this key material is presented too late.

6. The item auctioned off is a walk on a treadmill wearing an exoskeleton. MTMR focus on the case where this is a “bad” (rather than a “good), i.e., bidders are willing to pay in order NOT to have to walk. This is an awkward feature of the design! It rhymes badly with the introduction of the paper, where one is lead to believe that exoskeletons tend to have positive value to users, rather than being a drag on them. That said, I think one can live with this. I already commented on how awkward and contrived the “walk” is, so what difference does it make if the value is negative or positive? However, I would like to encourage MTMR discuss all the questionable aspects that I have brought up more candidly, as doing so will likely help readability and facilitate understanding.

7. Related to comment #6, I can well imagine that some users actually attach positive value to the walk, at least the first time they try it. I would be curious and intrigued to try an exoskeleton myself! How did MTMR deal with this possibility? They may say something somewhere in the paper, but in my view this is not discussed clearly enough.

8. Note that the value MTMR measure concerns a difference. Namely it is the difference between the values of (i) doing nothing and (ii) walking with an exoskeleton. For figuring out the value of exoskeleton use, this sees to be a non-obvious comparison. An alternative, possibly more natural, comparison could have involved (i’) walking without an exoskeleton and (ii’) walking with an exoskeleton. I propose that MTMR should flag for and discuss the issue.

9. MTMR say that they use SPA. However, the actual design, with automated co-players, in fact more closely resembles a device known in the experimental economics literature as a “Becker-DeGroot-Marschak (BDM) mechanism.” This needs to be acknowledged. In this connection one also needs to show awareness that some studies have indicated that subjects interacting with BDM-mechanisms often fail to rely on their dominant strategies. To the extent that this is true, it may undermine the usefulness of MTMR’ method. I do not think this observation damns the exercise; however, the concern should be noted. I recommend MTMR to look up and cite the reference Bohm, Linden & Sonnegard (1997, Economic Journal), and possibly also other references that cite that study.

10. In my view, introductions are more compelling (and papers overall more exciting) when no data is discussed at all. Rather, focus should be on describing and motivating the exercise from an ex ante point of view. The authors may want to consider adopting such an approach.

11. Many good movies involve exoskeletal action, like Edge of Tomorrow (with Tom Cruise) and some Spiderman movies. I wonder if the paper can be spiced up a bit with some cool references?

Reviewer #2 (Remarks to the Author):

This paper presents a unique approach for evaluating the effectiveness of exoskeletons through the introduction of a Vickrey auction format to determine the value that each user places on the exoskeleton's assistance. The author's found that the user's placed a moderate value on the assistance, but with a high degree of variability.

This is certainly an interesting idea. it is nice to see a new attempt at evaluation of exoskeletons, as the common method of metabolic benefit does seem to be missing many less tangible elements. To that end, however, the authors don't seem to spend much time on the less certain aspects of this auction. Specifically, it is a "gross" evaluation, so you can't isolate specific parts. Additionally, it seems like they are convinced that the Vickrey auction cannot be gamed by the subjects, which seems a little naive, as it is a subjective measure. I think that more attention should be spent presenting this as an alternative measure that we don't necessarily fully understand, rather than a great new alternative we can definitely use. In that respect, the paper is great - thinking outside the box on how to evaluate exoskeletons is critical, because we are clearly missing something as a field. However, we should be careful when presenting very subjective alternatives, and accept their shortcomings.

More high level feedback is:

A general challenge in reading the manuscript is determining the efficacy of the Vickrey auction as an evaluation tool. The actual mechanics of it are not presented until much later in the text, so the author's mostly have to "take the author's word" that it's a good method. While this is relatively similar to the standard layout of similar papers, this manuscript specifically may be improved by including a brief explanation of benefits of the Vickrey auction system ahead of time. For example, this reviewer was left wondering what the actual incentives for the subjects were to accurately bid, which was distracting when trying to read the results section.

Additionally, I am not sure I understand the differentials presented. The authors state that the value of the assistance was \$3.40, while the total value was \$19.76 and the cost of wearing is \$18.59. My expectation would be that difference between the cost and the value was this differential of \$3.40, but it is not. While I'm sure that's a misunderstanding on my part, it would be appreciated if the authors elaborated on how these measures were determined.

In the introduction, the authors talk about in lines 57-67 the benefits and evaluation of exoskeletons, and how they can be imperceptible to the user unless they have huge assistance. I think that this is somewhat incorrect. Often, the assistance becomes very evident when it stops. Additionally, I would say it is somewhat determined by the incentives and benefits of the device. For example, I don't think the benefits of the Nike fancy marathon shoes are directly evident during use given only a brief snapshot of time, but are evident when athletes view their performance increase over periods of time. In this way, if the correct measure and correct "incentive" structure is provided, I think exoskeleton user's may adopt the device. I do concede that this is somewhat the point of this paper, however, which is why I like it!

More specific feedback is:

I'm not convinced that reducing muscle activation necessarily reduces joint loading, as claimed on lines 30-31. Specifically, if the device is a soft exo-suit, the loads are still being carried through the joints, and are actually greater because they offset the weight of the device.

On lines 37-38, the claim that rehabilitation devices have a clear focus of repairing gait function is also not explicitly correct. There's many examples of attempting to use exoskeletons as a form of exercise (see work from Dr. Ann Spungen), which is subtly different from trying to restore function.

When comparing metabolic cost to the MV, that's a hard comparison, because metabolic cost is objective, while MV is extremely subjective. To the author's point, however, adoption of exoskeletons without the correct benefit structure will be inherently

subjective.

Please choose a different color scheme for the results. The green and orange are impossible to tell apart for colorblind people.

I'm not sure some of the claims are actually possible given the MV mean and standard deviation (6% vs 30%). This absolutely seems to be outside the tolerance of what can be said to be conclusive.

The code the authors reference is still listed as TBD.

The numbers listed in the methods section line 294 are not accurate. Firstly, the table only has 4 female participants, while they list 5 in the text, and there are 12 male participants, while the text lists 9.

How did the authors tune the robot bidders? It seems like this tuning has an enormous effect on the results, which isn't really mentioned.

On lines 432 and 433, what is MEV?

In Table 1, participant 8, the height is wrong.

In Table 2 and Table 3, it looks like participant 7/8's numbers got mixed up.

Dear editors and reviewers,

Thank you for your thoughtful and positive assessment of our work, which we have addressed below.

Reviewer 1

Comment 1.1 In second-price auctions (SPA), bidders have dominant strategies to reveal their valuations for the object auctioned off. If one is willing to assume that bidders rely on such strategies, bids will reveal valuations. For example, if a car is auctioned off under SPA rules, then submitted bids will reveal bidders' valuations. MTMR argue, compellingly in my view, that it may be useful to figure out how much value users of exoskeletons attach to such devices. Moreover, SPA may provide useful measuring rods in this connection. If it is the case the engineers have never thoughts along these lines before, then this message seems pioneering and worth having. I then support publication of a version of the paper. However, the manuscript, as currently formulated, has some serious flaws which need to be corrected. My comments below are meant to help MTMR fix the problems:

SPA are useful as valuation measuring rods only if it is interesting to learn the value of the item auctioned off. That might clearly be the case in my above example with a car, say if one is a company like Toyota and interested in figuring out demand for one's vehicles. It is furthermore compelling, as I said above, that someone may want to know the value that users of exoskeletons attach to such devices, in certain situations. However, I very much doubt that the situations studied in MTMR's experiment belongs to that class. Subjects buy (or sell) the right (or obligation) to take a "walk" on a treadmill wearing an exoskeleton. That seems to be a rather uninteresting or irrelevant item to buy or sell. For example, in the movie Edge of Tomorrow, Tom Cruise's character uses exoskeletons to fight off alien "Mimics," and it is clear that this is quite valuable to him. However, I doubt that he would have wanted to walk on treadmills instead. Accordingly, I feel that this study does not really produce any insights as regards the actual value of the use of exoskeletons in meaningful situations. In there is a contribution, it will rather concern some kind of proof-of-concept. The study alerts readers to the possibility of using SPA for evaluating engineered items, but there is no insight beyond that. The manuscript needs to be re-written to reflect this outlook.

Reply We thank the reviewer for recognizing the novelty of our approach. We are aware that the fact that our study occurred on a treadmill is a limitation that is relevant to the extensibility of our results to more realistic scenarios, where individuals will not be continuously walking uphill on a treadmill. However, despite this limitation, we believe our study still provides insight into how humans value exoskeletons and their assistance.

To provide some context about the state-of-the-art in exoskeleton research, the impact of exoskeletons is evaluated in terms of metabolic benefit. Moreover, design improvements to exoskeletons are generally conceived in terms of expected improvements to metabolic benefit. This approach thus emphasizes a singular "objective" benefit without considering important ways that this objective benefit might deviate from experience. One important way they could differ is in terms of discomfort. There is no metabolic measure that can account for how it feels to wear the exo. A second source of difference is that there may be threshold effects. People may notice metabolic benefits only at certain thresholds, as demonstrated in past work [1]. A key contribution of our work is finding a way to plot subjective experience, which we measure with the Vickrey SPA, in a context where other measures (e.g. metabolic benefit) are already understood. Thus, our goal is not necessarily to value an exo for a realistic scenario (which we agree with the reviewer this is not) but to better understand the sensitivities of subjective interpretations of value.

In terms of the "Edge of Tomorrow" scenario, this is a kind of test that might be used in a world in which experience with exoskeletons is already commonplace. However, in our current world, most people do not have any "priors" about what it feels like to use an exoskeleton, or how it would enhance or hinder their performance, and thus they cannot yet meaningfully attach a value to such a use case. Our controlled task of uphill walking gives them direct experience with the exoskeleton and then asks them to bid on further access to the device in the same context (walking more), thus insuring their bid is relatively informed. To do the Edge of Tomorrow type task with an SPA, we would first have to have subjects attempt this task with an exoskeleton a few times to get a sense of how it can help them.

In other words, the controlled nature of the task is needed here because we must assume that our subject pool has very limited experience with exoskeletons, so we need to provide them with that experience in the experiment and have that experience be comparable to an experience they understand

outside the experiment. Walking serves this goal.

Discussion, Paragraph 7 Additionally, although the task of uphill walking does not represent all possible uses for augmentative exoskeletons, it enables an opportunity to quantify value provided by the exoskeleton during an intuitive application where it can provide substantial benefit.

We intend for researchers to use our results to calculate the value of other exoskeleton devices during different tasks, and we now discuss this in the Discussion:

Discussion, Paragraph 7 Future studies could employ the Vickrey auction protocol to quantify the value of various different exoskeletons during different, more realistic use cases to gain a better understanding of when and which exoskeletons are perceived as valuable by users.

.....

Comment 1.2 Presumably, most readers of this journal do not know about SPA and their relevance for measuring valuations. A key objective of this paper should therefore be to clearly explain the method. However, the presentation on p4 is not very clear, including as regards conveying/proving the central insight that bidders have dominant strategies to equate their bids to their values. The presentation on p14 is better, but too relegated. Having both presentations is unnecessary and repetitive.

Reply Based on this feedback and the readers' likely unfamiliarity with the Vickrey auction concept, we have now substituted in the Vickrey auction explanation from the methods section to the main body of the paper.

Background on the Vickrey auction The Vickrey auction [2] is a powerful economic tool for determining the true value placed on goods or actions. In this type of auction, participants compete to purchase (or sell) a good or item. For each participant, the auction's structure is designed such that the optimal strategy for obtaining the item is to truthfully represent their internal value with their bid (e.g. to bid an amount equal to the true worth of the item). This optimality stems from the *second-price* nature of the auction [2, 3, 4], in which the winner is the participant who bids the highest (or lowest, as in the selling implementation used in this study).

Background on the Vickrey auction (cont'd) However, rather than paying the highest bid, the winner of the auction instead pays the *second-highest* bid. This feature—awarding the item to the highest bidder but requiring them to pay only the second highest bid—addresses a problem present in standard auctions in economic theory, which is that rational bidders will bid not only based on their own valuation of the item (the information the auction aims to reveal) but also their assessment of the other bidders' valuation of the item. Specifically, in a standard auction in which the item is awarded to the highest bidder at the price they bid, bidders have an incentive to bid only slightly above what they think the highest bid from their competitors will be, thus "under-bidding" their true value. The Vickrey auction removes this incentive. Participants do not know the value of competing bids (sealed-bid) before submitting their bids. In theory, bidders are disincentivized to bid less than their true value, as they run the risk of not winning the auction (not acquiring the good), and do not gain by bidding just above the second highest bid (since they pay that second highest bid in any event). Similarly, participants should not bid more than their true value, which otherwise could cause them to pay more than the value of the item. Thus, the second-price nature breaks the link between the auction winner and their specific bid. The inverse is also true for the case of selling an item (second-lowest bid is paid, lowest bid wins); thus, the incentive structure that elicits truthful bidding still holds in the seller's auction. Due to the presence of this optimal strategy, the Vickrey auction provides a method for quantifying the value of arbitrary goods, services, or abstract concepts [4, 5]. Prior researchers have also used Vickrey auction metrics to measure the value of abstract concepts or actions, such as food safety [6], GMO-free foods [7], the stigma resulting from HIV [8], personally identifiable information [9], and smartphone battery life [10]. In particular, Coursey *et al.* employed the Vickrey auction to quantify the willingness of participants to endure performing an unpleasant task, such as tasting a bitter liquid [11]. In our protocol, we use the Vickrey auction sequentially to repeatedly sample individuals' valuations of their time during uphill walking. Participants competed in a series of auctions in which they auctioned off their walking time for two-minute intervals in exchange for actual monetary compensation. If the participant won the auction, they accrued the payout and walked for the two-minute interval, whereas if they lost, they did not receive the payout and rested until the next interval. The participants bids across the sequential auctions denotes the wearer's "price-to-walk" curve. Participants walked in different walking conditions, which included normal unassisted walking and walking with exoskeleton augmentation; by comparing the price-to-walk curves for these different conditions, the value of the exoskeleton assistance can be extracted.

.....

Comment 1.3 On p4 MTMR explain how SPA is a “sealed-bid” auction, which is true. However, MTMR get it wrong what is the defining characteristic of that property. Namely, it is NOT that “bids are not revealed publicly.” Rather, it is that no bids are revealed to any bidder before that bidder submits his own bid. That is, others bids are sealed to a bidder when he submit his own bid. MTMR, by contrast, write as if it is critical that bidders are not told each others’ bids after the auction is over, which is not the case.

Reply We thank the reviewer for mentioning this clarification. While we know that traditional sealed bid auctions can have all bids revealed to the participants afterwards, in our case, we made the decision to only reveal the winning bid in an effort to prevent the subjects from being overly influenced by divulged information about the bids.

We now explicitly mention the traditional method in the Background section :

Background on the Vickrey auction Participants do not know the value of competing bids (sealed-bid) before submitting their bids.

We also now contrast and justify our method to reveal only the winning bid after each round of auctions:

Walking Protocol, Methods, Paragraph 2 At the end of each interval, the winning bid (second-lowest bid) was revealed; while in a traditional Vickrey second-price auction, all bids would be revealed, the participant was only given the winning bid to avoid unduly influencing them toward a specific bid range or incentivizing the participant to guess a pattern of the competing bids.

.....

Comment 1.4 MTMR’s exercise can be understood only if the SPA that is used is actually described. One key aspect is the item being auctioned off. What is it? Most awkwardly, MTMR never explain this in the main text. One has to flip back to p12+ to learn about this. The paper cannot be understood unless

the auction details are described, and, of course, this material has to be presented in the main text for the paper to be readable.

Reply We now explicitly discuss in the background section that participants competed to sell auction off their walking time.

Background on the Vickrey auction Participants competed in a series of auctions in which they auctioned off their walking time for two-minute intervals in exchange for actual monetary compensation.

.....

Comment 1.5 As I read, for a long time it wasn't clear to me that MTMR ran any auction at all, in the sense that if someone did not win the auction then they didn't get the item being auctioned off. Without that key feature, the whole study would have been flawed, and it would have been necessary to reject the submission. Now, MTMR did not make such a mistake. However, this is not revealed until p16, in the caption under Figure 7 ("should they lose, they instead. . ."). Overall, the presentation is obscure as this key material is presented too late.

Reply Thank you for pointing out this critical detail. We now explicitly mention this detail in the initial background paragraph on the Vickrey auction.

Background on the Vickrey auction If the participant won the auction, they accrued the payout and walked for the two-minute interval, whereas if they lost, they did not receive the payout and rested until the next interval.

.....

Comment 1.6 The item auctioned off is a walk on a treadmill wearing an exoskeleton. MTMR focus on the case where this is a "bad" (rather than a "good), i.e., bidders are willing to pay in order NOT to have to walk. This is an awkward feature of the design! It rhymes badly with the introduction of the paper, where one is lead to believe that exoskeletons tend to have positive value to users, rather than being a drag on them. That said, I think one can live with this. I already commented on how awkward and

contrived the “walk” is, so what difference does it make if the value is negative or positive? However, I would like to encourage MTMR discuss all the questionable aspects that I have brought up more candidly, as doing so will likely help readability and facilitate understanding.

Reply We thank the reviewer for this insight. We note that, although the task is strenuous and thus the user will naturally want to avoid it, because we are *comparing* their bids/values across these strenuous conditions, the value of the exoskeleton emerges from the comparison, *i.e.* the difference between participant price-to-walk for the different conditions. This comparison also allows for the exoskeleton to have either a positive or negative economic value, as we wanted to consider either case despite exoskeletons tending to have a net physiological benefit. We now highlight this comparison in the introduction section.

Introduction, Paragraph 5 We leveraged the Vickrey second-price auction to measure participants’ “price to walk,” across different walking conditions, such as walking with and without exoskeleton assistance. We obtain our economic value metric (*i.e.* MV) that captures the value of the exoskeleton by calculating the difference in cumulative value between these conditions. Since both cumulative values encode the baseline valuation of walking time, the difference in values is due to the effects of wearing the exoskeleton.

.....

Comment 1.7 Related to comment #6, I can well imagine that some users actually attach positive value to the walk, at least the first time they try it. I would be curious and intrigued to try an exoskeleton myself! How did MTMR deal with this possibility? They may say something somewhere in the paper, but in my view this is not discussed clearly enough.

Reply We now discuss that users’ evaluations of the exoskeleton may have been driven by an initial novelty effect, and that this study only sought to establish an initial investigation into the value of

exoskeletons.

Discussion, Paragraph 9 Similarly, it is possible that the initial MV may have been partly driven by the positive value a user places on the novelty of wearing an exoskeleton, if the experience was novel for them. In our protocol, to reduce the influence of this possibility, we allowed the participants to complete an adaptation period in which they freely walked with the exoskeleton prior to undergoing the full experiment. For this initial investigation, we sought to understand the immediate economic value obtained from a first-time experience of exoskeleton use, analogous to a user assessing an exoskeleton when making the decision to adopt (*i.e.* a “test drive”).

.....

Comment 1.8 Note that the value MTMR measure concerns a difference. Namely it is the difference between the values of (i) doing nothing and (ii) walking with an exoskeleton. For figuring out the value of exoskeleton use, this seems to be a non-obvious comparison. An alternative, possibly more natural, comparison could have involved (i') walking without an exoskeleton and (ii') walking with an exoskeleton. I propose that MTMR should flag for and discuss the issue.

Reply We acknowledge that a comparison between walking with and without the exoskeleton between sequential trials appears more intuitive. However, it can be logistically difficult to don and doff the exo in the short trial time span we chose, which as explained in our paper, was driven by factors such as experimental time and the fatiguing effect of the locomotion task. We note that in our study, we make the comparison suggested by the reviewer by calculating MV using the difference between the cumulative prices of the different walking trials, which isolates the effects of the exoskeleton (see Analysis section

under Methods). We now clarify the nature of these comparisons in the Results section

Discussion, Paragraph 2 We obtained these results by quantifying the marginal value (MV) between the different conditions, which captured the economic benefit or detriment between these conditions. To isolate the net value of the exoskeleton + assistance, we directly calculated the difference in value between unassisted walking, which acts as a baseline control condition, and exoskeleton-assisted walking, which introduces the economic value provided by the exoskeleton. To calculate the cost of wearing the unpowered exoskeleton, we compared the values of unassisted walking to that of walking with the unpowered device. The value of the assistance alone was then isolated by comparing the price-to-walk curves of the exoskeleton-assisted condition and the unpowered condition, which controlled for the value of the participants' time and thus enabled assessment of the detriment caused by the device's mass.

.....

Comment 1.9 MTMR say that they use SPA. However, the actual design, with automated co-players, in fact more closely resembles a device known in the experimental economics literature as a "Becker-DeGroot-Marschak (BDM)mechanism." This needs to be acknowledged. In this connection one also needs to show awareness that some studies have indicated that subjects interacting with BDM-mechanisms often fail to rely on their dominant strategies. To the extent that this is true, it may undermine the usefulness of MTMR' method. I do not think this observation damns the exercise; however, the concern should be noted. I recommend MTMR to look up and cite the reference Bohm, Linden & Sonnegard (1997, Economic Journal), and possibly also other references that cite that study.

Reply We thank the reviewer for bringing this paper to our attention. We now discuss the similarities (and key differences) between our study and the work by Bohm, Linden & Sonnegard in the discussion

section.

Discussion, Paragraph 11 The presence of automated robo-bidders, rather than actual humans, rendered our protocol similar to an auction that used the Becker-DeGroot-Marschak (BDM) method [12], in which participants compete in auctions against bids generated randomly via a statistical distribution, rather than other humans. It has been shown that the BDM auction is incentive compatible with the strategy of truthful bidding to sell an item (as in the Vickrey auction) when the maximum buyout price—the maximum the seller could be expected to receive—generated by the distribution does not exceed a realistic buyout price for the good and when the sellers are aware of this concept [13]. However, we note that unlike in typical BDM auctions, our approach features repeated Vickrey auctions with realistic competing bids, rather than stochastically-generated bids. Thus, after each auction when the winning bid was disclosed, subjects had a general understanding of the range of bids that would be expected in the experiment. Additionally, even if the participant bids given were greater than honest bids, the net MVs calculated would be similar due to the MV metric being calculated using the difference of cumulative prices (provided the magnitude of the effect did not vary across days). A future study that replicates the protocols in this work while including all human participants would be informative.

.....

Comment 1.10 In my view, introductions are more compelling (and papers overall more exciting) when no data is discussed at all. Rather, focus should be on describing and motivating the exercise from an ex ante point of view. The authors may want to consider adopting such an approach.

Reply We appreciate the reviewer’s suggestion. However, we defer to the norms of our field, which typically provide some quantifiable data in the introduction to strengthen the motivation for the study. For example, we specifically mention the reductions in metabolic rate provided by modern exoskeletons to highlight the great strides made by the field in improving the efficiency in walking, and to underscore that, despite the primacy of the metabolic rate metric, it may not capture true usefulness to the wearer.

.....

Comment 1.11 Many good movies involve exoskeletal action, like Edge of Tomorrow (with Tom Cruise) and some Spiderman movies. I wonder if the paper can be spiced up a bit with some cool

references?

Reply We have changed the introduction to incorporate an allusion to popular culture in the public consciousness.

Introduction, Paragraph 1 Powered exoskeletons have long fascinated the public consciousness with their promise to supersede the limitations of human performance. In parallel, there has been substantial growth of scientific research into powered lower-limb exoskeletons, driven by the potential of these technologies to transform mobility by extending the locomotor abilities of their wearers.

.....

Reviewer 2

Comment 2.1 This paper presents a unique approach for evaluating the effectiveness of exoskeletons through the introduction of a Vickrey auction format to determine the value that each user places on the exoskeleton's assistance. The author's found that the user's placed a moderate value on the assistance, but with a high degree of variability.

This is certainly an interesting idea. it is nice to see a new attempt at evaluation of exoskeletons, as the common method of metabolic benefit does seem to be missing many less tangible elements. To that end, however, the authors don't seem to spend much time on the less certain aspects of this auction. Specifically, it is a "gross" evaluation, so you can't isolate specific parts. Additionally, it seems like they are convinced that the Vickrey auction cannot be gamed by the subjects, which seems a little naive, as it is a subjective measure. I think that more attention should be spent presenting this as an alternative measure that we don't necessarily fully understand, rather than a great new alternative we can definitely use. In that respect, the paper is great - thinking outside the box on how to evaluate exoskeletons is critical, because we are clearly missing something as a field. However, we should be careful when presenting very subjective alternatives, and accept their shortcomings.

Reply We thank the reviewer for this feedback and for the support for new metrics. We now clarify that, results aside, this study is meant to be an initial investigation into using the economic value metrics

as an alternative to the metric of metabolic cost reduction.

Introduction, Paragraph 5 Thus, this study represents an initial investigation into the use of economic value as a metric to assess the success of exoskeletons and their assistance; we believe assessment of economic value represents a potentially useful alternative to the dominant approach of quantifying the reduction in metabolic rate.

We also discuss that our study identifies several areas for future work that would shed light on the Vickrey auction as a tool to quantify the value of different exoskeletons, such as any potential effect of different robo-bidder parameters and different exoskeleton use cases, as well as the effects of having computerized bidders altogether. These discussions also now include acknowledgements of the potential for subjects to not bid truthfully as the dominant strategy.

Discussion, Paragraph 10 Any series of auctions would naturally establish an equilibrium between the participants; thus, by using robo-bidders, we were able to control for this equilibrium, which strengthens our ability to compare across subjects. Although we expect that, given honest bidding, participant values will remain constant, future work should investigate the effect of changing robo-bidder parameters on participant bids. In addition, the use of robo-bidders enables our results to more readily be compared across other researchers, institutions, and exoskeletons that are assessed using comparable methods.

Discussion, Paragraph 7 Future studies could employ the Vickrey auction protocol to quantify the value of various different exoskeletons during different, more realistic use cases to gain a better understanding of when and which exoskeletons are perceived as valuable by users.

Discussion, Paragraph 11 Participants received the expected monetary compensation that resulted from their winning bids, which similarly incentivized truthfulness. The presence of automated robo-bidders, rather than actual humans, rendered our protocol similar to an auction that used the Becker-DeGroot-Marschak (BDM) method [12], in which participants compete in auctions against bids generated randomly via a statistical distribution, rather than other humans. It has been shown that the BDM auction is incentive compatible with the strategy of truthful bidding to sell an item (as in the Vickrey auction) when the maximum buyout price—the maximum the seller could be expected to receive—generated by the distribution does not exceed a realistic buyout price for the good and when the sellers are aware of this concept [13]. However, we note that unlike in typical BDM auctions, our approach features repeated Vickrey auctions with realistic competing bids, rather than stochastically-generated bids. Thus, after each auction when the winning bid was disclosed, subjects had a general understanding of the range of bids that would be expected in the experiment. Additionally, even if the participant bids given were greater than honest bids, the net MVs calculated would be similar due to the MV metric being calculated using the difference of cumulative prices (provided the magnitude of the effect did not vary across days). A future study that replicates the protocols in this work while including all human participants would be informative.

.....

Comment 2.2 A general challenge in reading the manuscript is determining the efficacy of the Vickrey auction as an evaluation tool. The actual mechanics of it are not presented until much later in the text, so the author's mostly have to "take the author's word" that it's a good method. While this is relatively similar to the standard layout of similar papers, this manuscript specifically may be improved by including a brief explanation of benefits of the Vickrey auction system ahead of time. For example, this reviewer was left wondering what the actual incentives for the subjects were to accurately bid, which was distracting when trying to read the results section.

Reply Based on this feedback and the readers' likely unfamiliarity with the Vickrey auction concept, we have bolstered the explanation for the Vickrey auction in the main body of the paper.

Background on the Vickrey auction The Vickrey auction [2] is a powerful economic tool for determining the true value placed on goods or actions. In this type of auction, participants compete to purchase (or sell) a good or item. For each participant, the auction's structure is designed such that the optimal strategy for obtaining the item is to truthfully represent their internal value with their bid (e.g. to bid an amount equal to the true worth of the item). This optimality stems from the *second-price* nature of the auction [2, 3, 4], in which the winner is the participant who bids the highest (or lowest, as in the selling implementation used in this study). However, rather than paying the highest bid, the winner of the auction instead pays the *second-highest* bid. This feature—awarding the item to the highest bidder but requiring them to pay only the second highest bid—addresses a problem present in standard auctions in economic theory, which is that rational bidders will bid not only based on their own valuation of the item (the information the auction aims to reveal) but also their assessment of the other bidders' valuation of the item. Specifically, in a standard auction in which the item is awarded to the highest bidder at the price they bid, bidders have an incentive to bid only slightly above what they think the highest bid from their competitors will be, thus "under-bidding" their true value. The Vickrey auction removes this incentive. Participants do not know the value of competing bids (sealed-bid) before submitting their bids. In theory, bidders are disincentivized to bid less than their true value, as they run the risk of not winning the auction (not acquiring the good), and do not gain by bidding just above the second highest bid (since they pay that second highest bid in any event). Similarly, participants should not bid more than their true value, which otherwise could cause them to pay more than the value of the item. Thus, the second-price nature breaks the link between the auction winner and their specific bid. The inverse is also true for the case of selling an item (second-lowest bid is paid, lowest bid wins); thus, the incentive structure that elicits truthful bidding still holds in the seller's auction. Due to the presence of this optimal strategy, the Vickrey auction provides a method for quantifying the value of arbitrary goods, services, or abstract concepts [4, 5]. Prior researchers have also used Vickrey auction metrics to measure the value of abstract concepts or actions, such as food safety [6], GMO-free foods [7], the stigma resulting from HIV [8], personally identifiable information [9], and smartphone battery life [10].

Background on the Vickrey auction (cont'd) In particular, Coursey *et al.* employed the Vickrey auction to quantify the willingness of participants to endure performing an unpleasant task, such as tasting a bitter liquid [11]. In our protocol, we use the Vickrey auction sequentially to repeatedly sample individuals' valuations of their time during uphill walking. Participants competed in a series of auctions in which they auctioned off their walking time for two-minute intervals in exchange for actual monetary compensation. If the participant won the auction, they accrued the payout and walked for the two-minute interval, whereas if they lost, they did not receive the payout and rested until the next interval. The participants bids across the sequential auctions denotes the wearer's "price-to-walk" curve. Participants walked in different walking conditions, which included normal unassisted walking and walking with exoskeleton augmentation; by comparing the price-to-walk curves for these different conditions, the value of the exoskeleton assistance can be extracted.

.....

Comment 2.3 Additionally, I am not sure I understand the differentials presented. The authors state that the value of the assistance was \$3.40, while the total value was \$19.76 and the cost of wearing is \$18.59. My expectation would be that difference between the cost and the value was this differential of \$3.40, but it is not. While I'm sure that's a misunderstanding on my part, it would be appreciated if the authors elaborated on how these measures were determined.

Reply We thank the review for pointing this out. The reason the dollar values are not a straightforward subtraction is because the dollar values are calculated by multiplying the MVs for each condition by the average cumulative price. While the average cumulative price was calculated using every subject's data, the MVs for each condition were not necessarily calculated using every subject's data; for example, not every participant experienced the unpowered exoskeleton condition, so only those who did were used

to calculate the unpowered MV. We now incorporate this analysis into the discussion.

Discussion, Paragraph 2 The value of each condition in dollars/hr was obtained by expressing the average cumulative price as a rate of dollars/hr for the unassisted condition, and multiplying this rate by the MV of each condition, expressed in percentage change from the unassisted condition. The resulting hourly rates therefore represent the change in dollar value from the baseline unassisted condition. While every subject completed at least the unassisted condition and the powered exoskeleton condition, not every participant completed the unpowered condition, and thus we cannot simply take the difference between the average value of the assistance and the average penalty of the exoskeleton's mass to obtain the net value. The average cumulative price of walking uphill for 30 minutes without the exoskeleton was \$29.20 (\$58.50/hr SD: \$57.50/hr). The MV of wearing the unpowered exoskeleton—that was not providing assistance—was -31.8%, which translates to a monetary cost of \$18.60/hr (SD: \$18.30/hr) for wearing the unpowered system; this value in dollars was calculated by multiplying the average cumulative price per hour (\$58.50/hr) and the MV of the unpowered exoskeleton (-31.8%, N=10). When assistance was applied by the exoskeleton, the MV increased to just above zero (\$3.40/hr, SD:\$3.40/hr) when compared to not wearing the exoskeleton, which was calculated by multiplying the average cumulative price by the MV of exoskeleton use (5.8%, N=16). The marginal value added by the assistance alone was 33.8%, which translates to an added value of \$19.80/hr (SD: \$19.40/hr); which was calculated by multiplying the average cumulative price by the MV for the assistance alone (33.8%, N=10).

.....

Comment 2.4 In the introduction, the authors talk about in lines 57-67 the benefits and evaluation of exoskeletons, and how they can be imperceptible to the user unless they have huge assistance. I think that this is somewhat incorrect. Often, the assistance becomes very evident when it stops. Additionally, I would say it is somewhat determined by the incentives and benefits of the device. For example, I don't think the benefits of the Nike fancy marathon shoes are directly evident during use given only a brief snapshot of time, but are evident when athletes view their performance increase over periods of time. In this way, if the correct measure and correct "incentive" structure is provided, I think exoskeleton user's may adopt the device. I do concede that this is somewhat the point of this paper, however, which is why I like it!

Reply We thank the reviewer for this comment and their praise for the work. The reviewer is right to remind us of this argument’s restriction to short-term valuations. Our view is informed by our own prior work characterizing this human perception in the short term; we now clarify the temporal scope of the claim.

Introduction, Paragraph 3 Our recent work has shown that during short-term exoskeleton-assisted walking, the average user cannot yet perceive the benefit of most systems available today [1, 14, 15].

Introduction, Paragraph 7 In this study, we introduce a tool for measuring the perceived short term economic value of exoskeleton use as a metric to evaluate their performance and user experience.

Discussion, Paragraph 1 In this work, we use the Vickrey second-price auction as a method to capture the economic value or detriment provided by exoskeletons and their assistance in the immediate term.

Discussion, Paragraph 9 Similarly, it is possible that the initial MV may be have been partly driven by the positive value a user places on the novelty of wearing an exoskeleton, if the experience was novel for them. In our protocol, to reduce the influence of this possibility, we allowed the participants to complete an adaptation period in which they freely walked with the exoskeleton prior to undergoing the full experiment. For this initial investigation, we sought to understand the immediate economic value obtained from a first-time experience of exoskeleton use, analogous to a user assessing an exoskeleton when making the decision to adopt (i.e. a “test drive”). Future work is needed to understand any adaptation of value that may occur over time, which would have implications in the longer-term value of these technologies.

As the reviewer points out, the goal of this study was to characterize incentives to use exoskeletons as encoded by our metric of economic value.

.....

Comment 2.5 I’m not convinced that reducing muscle activation necessarily reduces joint loading, as claimed on lines 30-31. Specifically, if the device is a soft exo-suit, the loads are still being carried through the joints, and are actually greater because they offset the weight of the device.

Reply We thank the reviewer for pointing this out. Prior work we have conducted has investigated precisely the effect of soft exosuits on joint loading, which corroborates the reviewer’s comment. We now acknowledge this work in the introduction.

Introduction, Paragraph 1 In addition, exoskeletons can reduce muscle activation, and thus certain exoskeleton architectures may reduce joint loading [16, 17, 18, 19], potentially extending the physical capabilities of aging individuals.

.....

Comment 2.6 On lines 37-38, the claim that rehabilitation devices have a clear focus of repairing gait function is also not explicitly correct. There’s many examples of attempting to use exoskeletons as a form of exercise (see work from Dr. Ann Spungen), which is subtly different from trying to restore function.

Reply We have rewritten the introduction to distinguish this related but different application of rehabilitative exoskeletons, and have included citations to Dr. Spungen’s work.

Introduction, Paragraph 2 Rehabilitative exoskeletons can achieve this goal by directly affecting the kinetics and kinematics via their applied assistance [20, 18, 21, 22, 23] or by using the exoskeleton as a training aid to foster neurorehabilitation [24, 25, 26]. Thus, the use of these exoskeletons provides a clear physiological objective on which to base the design and control decisions required for development.

.....

Comment 2.7 When comparing metabolic cost to the MV, that’s a hard comparison, because metabolic cost is objective, while MV is extremely subjective. To the author’s point, however, adoption of exoskeletons without the correct benefit structure will be inherently subjective.

Reply We now discuss the need for future work to discover what, if any, link exists between metabolic cost and MV, and how this can inform the targeting of individuals for exoskeleton adoption.

Discussion, Paragraph 3 With respect to the current standard of metabolic cost reductions, the metric of MV and the notion of the price-to-walk more holistically quantify the experience of exoskeleton use, including subjective evaluations, to a greater degree than singular physiological objectives. Thus, our work motivates future investigations to discover the degree to which economic value encodes metabolic benefits of exoskeletons.

.....

Comment 2.8 Please choose a different color scheme for the results. The green and orange are impossible to tell apart for colorblind people.

Reply Based on this helpful recommendation (and the color palettes found here: <https://davidmathlogic.com/colorblind/>) we have replaced the green and orange with different shades designed to be accessible to colorblind people.

.....

Comment 2.9 I'm not sure some of the claims are actually possible given the MV mean and standard deviation (6% vs 30%). This absolutely seems to be outside the tolerance of what can be said to be conclusive.

Reply We note that despite the large standard deviations present, t-tests for the effects on value of the exoskeleton assistance alone and the unpowered exoskeleton are significant (see Results, Paragraph 2). It is only for the net MV (assistance plus device) that the p-value is not significant, and thus we state that we cannot distinguish the net value of exoskeletons from zero. We now include the Standard Error of the Mean (SEM) when reporting the statistics of the results, which is used when performing the

t-tests.

Results, Paragraph 2 The average inter-subject MV of exoskeleton use was 5.8%, with a standard deviation (SD) of 31.14% (N=16, SEM=7.8%, Fig. 2a). The exoskeleton + assistance thus provided only a small value benefit to the average participant. Using a t-test, the average MV of exoskeleton + assistance (5.8%) was not significantly different from zero ($p = 0.24$). However, as denoted by the high standard deviation, some participants received large benefits from the device's assistance, while others experienced an economic penalty from exoskeleton use. The average MV for the unpowered exoskeleton was -31.8% with a standard deviation of 45.0% (N=10, SEM=14.2%, 2b). Using the same t-test as in the exo-powered condition, we found this change to be significantly different from zero ($p = 0.03$). Additionally, the powered assistance alone from the exoskeleton provided a significant increase in value (mean: 33.8%, SD: 38.1%, SEM=12.0%, $p = 0.01$, N=10, Fig. 2c).

.....

Comment 2.10 The code the authors reference is still listed as TBD.

Reply Thank you for pointing this out. We have added in the citation to the CodeOcean capsule containing our code, available at the following link: <https://codeocean.com/capsule/7159524/tree/v1>

.....

Comment 2.11 The numbers listed in the methods section line 294 are not accurate. Firstly, the table only has 4 female participants, while they list 5 in the text, and there are 12 male participants, while the text lists 9.

Reply Thank you for pointing out this inconsistency. We have updated the numbers in the text to accurately reflect the numbers in the table.

Participants, Methods, Paragraph 1 In this study, sixteen able-bodied participants (N = 16, 4 female, 12 male...

Comment 2.12 How did the authors tune the robot bidders? It seems like this tuning has an enormous effect on the results, which isn't really mentioned.

Reply We now explicitly discuss how the robot-bidder behavior was created.

Robo-bidders, Methods, Paragraph 1 The mechanics of the robo-bidders are modeled on the bidding behavior of three pilot subjects, which featured an expected trend of increased bids as they walked for longer on the uphill walking task.

To the reviewer's second point, we note that, regardless of if the competing bids came from humans or robots, the participants' own bids must reach an equilibrium behavior with these competing bids that is naturally dictated by the participants' own values.

Discussion, Paragraph 10 The use of robo-bidders also enabled us to standardize the interaction between the human participant and the other auction participants. Any series of auctions would naturally establish an equilibrium between the participants; thus, by using robo-bidders, we were able to control for this equilibrium, which strengthens our ability to compare across subjects.

We now also discuss the need for future investigations to understand the effect of changing robo-bidder behavior on human participant bids.

Discussion, Paragraph 10 Although we expect that, given honest bidding, participant values will remain constant, future work should investigate the effect of changing robo-bidder parameters on participant bids.

.....

Comment 2.13 On lines 432 and 433, what is MEV?

Reply Thank you for pointing out this typo. MEV was an earlier acronym for MV, and we have accordingly replaced all mentions of MEV in the text with MV.

.....

Comment 2.14 In Table 1, participant 8, the height is wrong.

Reply We have updated the table with the correct height.

.....

Comment 2.15 In Table 2 and Table 3, it looks like participant 7/8's numbers got mixed up.

Reply We have updated the numbers in Tables 2 and 3 to fix this mistake.

.....

.....

Thank you for taking the time to provide all your comments and helping us improve this paper.

Sincerely,

The authors, Roberto Leo Medrano, Gray Cortright Thomas, Drew Margolin, and Elliott J. Rouse

References

- [1] R. L. Medrano, G. C. Thomas, E. J. Rouse, *Journal of NeuroEngineering and Rehabilitation* **19**, 1 (2022).
- [2] W. Vickrey, *The Journal of Finance* **16**, 8 (1961).
- [3] S. Parsons, J. A. Rodriguez-Aguilar, M. Klein, *ACM Computing Surveys (CSUR)* **43**, 1 (2011).
- [4] D. Lucking-Reiley, *Journal of Economic Perspectives* **14**, 183 (2000).
- [5] H. R. Varian, C. Harris, *American Economic Review* **104**, 442 (2014).
- [6] J. A. Fox, D. J. Hayes, J. F. Shogren, *Journal of risk and Uncertainty* **24**, 75 (2002).
- [7] M. C. Rousu, J. L. Lusk, *AgBioForum* **12**, 226 (2009).
- [8] V. Hoffmann, J. R. Fooks, K. D. Messer, *Economic Development and Cultural Change* **62**, 701 (2014).
- [9] J. Staiano, *et al.*, *UbiComp 2014 - Proceedings of the 2014 ACM International Joint Conference on Pervasive and Ubiquitous Computing* pp. 583–594 (2014).
- [10] S. Hosio, *et al.*, *Conference on Human Factors in Computing Systems - Proceedings* pp. 1869–1880 (2016).
- [11] D. Coursey, J. L. Hovis, W. D. Schulze **102**, 679 (2016).
- [12] G. M. Becker, M. H. DeGroot, J. Marschak, *Behavioral science* **9**, 226 (1964).
- [13] P. Bohm, J. Lindén, J. Sonnegård, *The Economic Journal* **107**, 1079 (1997).
- [14] R. L. Medrano, G. C. Thomas, E. Rouse, *Proceedings of the IEEE RAS and EMBS International Conference on Biomedical Robotics and Biomechatronics 2020-November*, 483 (2020).
- [15] R. L. Medrano, G. C. Thomas, E. Rouse, *Dynamic Walking 2018* .
- [16] T. Lenzi, M. C. Carrozza, S. K. Agrawal, *IEEE Transactions on Neural Systems and Rehabilitation Engineering* **21**, 938 (2013).

- [17] C. L. Lewis, D. P. Ferris, *Journal of Biomechanics* **44**, 789 (2011).
- [18] H. Zhu, C. Nesler, N. Divekar, M. Ahmad, R. Gregg, *IEEE Int. Conf. Rehab. Robot.* (2019).
- [19] R. L. Medrano, E. J. Rouse, G. C. Thomas, *IEEE Transactions on Medical Robotics and Bionics* pp. 1–4.
- [20] H. Zhu, *et al.*, *Proceedings - IEEE International Conference on Robotics and Automation* pp. 504–510 (2017).
- [21] S. Sridar, P. H. Nguyen, M. Zhu, Q. P. Lam, P. Polygerinos, *IEEE International Conference on Intelligent Robots and Systems 2017-Septe*, 3722 (2017).
- [22] B. Chen, B. Zi, Z. Wang, L. Qin, W. H. Liao, *Mechanism and Machine Theory* **134**, 499 (2019).
- [23] G. S. Sawicki, D. P. Ferris, *Journal of NeuroEngineering and Rehabilitation* **6**, 1 (2009).
- [24] A. Ramanujam, *et al.*, *2018 40th Annual International Conference of the IEEE Engineering in Medicine and Biology Society (EMBC)* (IEEE, 2018), pp. 2805–2808.
- [25] P. K. Asselin, M. Avedissian, S. Knezevic, S. Kornfeld, A. M. Spungen, *JoVE (Journal of Visualized Experiments)* p. e54071 (2016).
- [26] D. B. Fineberg, *et al.*, *The journal of spinal cord medicine* **36**, 313 (2013).

REVIEWERS' COMMENTS:

Reviewer #1 (Remarks to the Author):

The authors have responded well to my comments. The manuscript now looks great. I have no further comments. I support publication.

Reviewer #2 (Remarks to the Author):

The revisions provided by the authors address all the concerns that I have before this paper could be published.